# Harnessing *Wolbachia* cytoplasmic incompatibility alleles for confined gene drive: A modeling study

**Jiahe Li, Jackson Champer** [ID] *

Center for Bioinformatics, School of Life Sciences, Peking-Tsinghua Center for Life Sciences, Peking University, Beijing, China

* jchamper@pku.edu.cn

**Data Availability Statement:** All data is available in the manuscript, and All MATLAB and SLiM models are available at https://github.com/jchamper/ChamperLab/tree/main/CifAB-Drive-Modeling.

## Abstract

*Wolbachia* are maternally-inherited bacteria, which can spread rapidly in populations by manipulating reproduction. *cifA* and *cifB* are genes found in *Wolbachia* phage that are responsible for cytoplasmic incompatibility, the most common type of *Wolbachia* reproductive interference. In this phenomenon, no viable offspring are produced when a male with both *cifA* and *cifB* (or just *cifB* in some systems) mates with a female lacking *cifA*. Utilizing this feature, we propose new types of toxin-antidote gene drives that can be constructed with only these two genes in an insect genome, instead of the whole *Wolbachia* bacteria. By using both mathematical and simulation models, we found that a drive containing *cifA* and *cifB* together creates a confined drive with a moderate to high introduction threshold. When introduced separately, they act as a self-limiting drive. We observed that the performance of these drives is substantially influenced by various ecological parameters and drive characteristics. Extending our models to continuous space, we found that the drive individual release distribution has a critical impact on drive persistence. Our results suggest that these new types of drives based on *Wolbachia* transgenes are safe and flexible candidates for genetic modification of populations.

## Author summary

*Wolbachia* bacteria can be placed into insects and released into the wild, where the *Wolbachia* spreads throughout the insect population. They have been used as an effective disease control tool because the *Wolbachia* reduces transmission of pathogens by mosquitoes. It may also be possible to use the *Wolbachia* spread mechanism to power a gene drive, allowing for flexible deployment of cargo genes in insect populations. We used a mathematical reaction-diffusion model to assess the possibility of using *Wolbachia* phage genes *cifA* and *cifB* as a gene drive. In this system, these genes are inserted directly into the insect genome, and females will fail to have offspring with drive males unless they also have a drive allele. We model its characteristics under a range of performance parameters in both simple panmictic populations and continuous space populations. Overall, we find that our

**Funding:** This study was supported by laboratory startup funds from Peking University and the SLS-Qidong Innovation Fund to JC. The funders had no role in study design, data collection and analysis, decision to publish, or preparation of the manuscript.

**Competing interests:** The authors have declared that no competing interests exist.

CifAB drive can be highly confined to a target population, yet provides a powerful option for population modification of insect species.

## Introduction

### Gene drive systems

Gene drives are selfish genetic elements that increase their transmission to the next generation and spread at a higher rate than expected under Mendelian inheritance [1–6]. Engineered gene drive systems thus have a broad spectrum of potential applications and can be classified into two categories: suppression drives and modification drives. Suppression drives may be used to reduce or eliminate populations of pest species such as invasive fire ants, pesticide-resistant agricultural pests, or mosquito disease vectors. Modification drives can be used to propagate advantageous characteristics across populations faster than natural evolution or enable a trait to spread despite having a fitness cost. For example, they can be used to provide an allele to rescue an endangered species or a disease-refractory allele against malaria and dengue in mosquitoes, reducing the transmission of these diseases. Overall, gene drives have great potential to provide solutions to a variety of public health and environmental issues by rapidly spreading through wild populations.

In the past few years, considerable progress has been made in the field. Homing drives making use of the CRISPR system have proven efficient for both modification [7–9] and suppression [10–12], though resistance [13–17] and in some cases drive efficiency [12] remain as obstacles to their success. Various types of toxin-antidote drives have also been modeled [18–24]. Some of these have been proven to be capable of spreading in *Drosophila* experiments [25–31]. However, more research is needed to find suitable promoters and target sites and to successfully construct these new types of drives with desired properties in target species.

### Spatially confined gene drives

Spatially confined gene drives are alleles that need to be introduced into the population above a certain frequency to spread in the population [20, 32, 33]. If the drive introduction frequency is below the threshold, the drive allele will be eliminated from the population over time. When introduced above this frequency, the drive will further increase in frequency towards fixation or its equilibrium. Spatially confined drives are "local" drives, since when the drive occupies one of two demes that are linked together, a sufficiently low level of migration will stop the drive from ever exceeding its threshold in the other deme, preventing it from taking over the population. In contrast, "global" gene drives such as homing drives have an introduction threshold of zero and can spread rapidly once the drive allele is present in a population. Utilizing this property of spatially confined gene drives, they could be applied to scenarios where only regional genetic modification of the population is desired, such as cases where we attempt to eliminate an invasive species from a foreign habitat while preventing the gene drive from affecting the species in its native habitat. This type of drive could also be more desirable from a sociopolitical or biosafety standpoint.

Often, spatial confinement is enabled through underdominance. Underdominance is the phenomenon in which drive/wild-type heterozygotes have a lower chance of survival and reproduction than either drive or wild-type homozygotes. Several designs for underdominance gene drives have been proposed and modeled computationally [19, 30]. However, they are

difficult to engineer in pests of interest due to the need for specific promoters, highly specific target sequences, and RNAi elements, or high fitness costs [19, 34].

### Self-limiting drives

Self-limiting drives tend to spread in a population for a certain number of generations and then decline in frequency until they are eliminated [35, 36]. This provides a different sort of confinement than described above, explicitly limiting the drive in time, rather than space (though the ability of self-limiting drives to spread through spatially structured populations can also be reduced in some cases). The mechanism underlying these drives often involves a "killer-rescue system", in which the killer is a gene with toxic effects on its carriers, and the unlinked rescue gene can mitigate these effects [35, 37, 38]. As long as there is a fitness cost to the rescue allele, both the killer and rescue gene will eventually be lost from the population. Therefore, for self-limiting drives to persist in a population for a long time, continuous releases or very low fitness costs are necessary. Some of these drives have been experimentally demonstrated in flies [39–41]. In many cases, we only need the drive to spread in the population for a narrow window of time, making self-limiting drives potentially desirable. This can be when more stringent control of transgenic alleles in the wild is desired, for instance, or when attempting to temporarily modify the traits of a population for testing. These drives often require careful control of release sizes and efficiency to obtain the necessary balance of initial spread and the required level of persistence.

### *Wolbachia*

*Wolbachia* are maternally inherited bacteria that are found in many arthropod species. They can manipulate the reproduction of their hosts by several mechanisms, such as by distorting sex ratios, thus increasing their frequency in insect populations [42–47]. The most common form of manipulation is cytoplasmic incompatibility. This phenomenon leads to no viable offspring when a *Wolbachia*-infected male mates with an uninfected female. However, if the female is also infected, the offspring will be viable. Because *Wolbachia* is maternally transmitted, it can therefore act like a gene drive, rapidly spreading through insect populations despite a moderate fitness cost.

Attempts have been made to directly utilize *Wolbachia* for population manipulation in the field. Since wild-type females cannot have viable offspring with *Wolbachia*-infected males, releasing large amounts of infected males could reduce the number of offspring produced in the population since female mosquitoes usually only mate once. Therefore, some Sterile Insect Technique programs are *Wolbachia*-based. The release of *Wolbachia*-infected insects could serve as a powerful tool for controlling pests like *Drosophila suzukii* [48], *Aedes aegypti* [49, 50] and *Aedes albopictus* [51].

Furthermore, *Wolbachia* itself can reduce the transmission of diseases like dengue in *Aedes aegypti*, so it can be applied like a modification drive, with the *Wolbachia* spreading and persisting in the population after the initial release [46, 48, 51, 52]. However, introducing a bacteria can inflict a large fitness cost, which requires a higher introduction frequency for the drive to spread efficiently (the threshold of *Wolbachia* releases is zero in the ideal case, but a nonzero threshold appears with even small fitness costs). Further, it also creates a larger change in the population compared to a drive allele, with the introduced bacteria containing thousands of potentially novel genes and more unexpected interactions that a small transgene package of a gene drive. Nevertheless, large-scale *Wolbachia* releases have been quite effective at spread the bacteria and subsequently reducing dengue transmission. For instance, releases of *Aedes*

*aegypti* which were infected with the *Wolbachia* strain *w*Mel were able to reduce the spread of dengue in Cairns, Australia [52–54].

### Cytoplasmic incompatibility genes

A pair of genes commonly associated with phage WO, which infects *Wolbachia*, underlie the mechanism of this bacterium's powerful cytoplasmic incompatibility-based drive. Transgenic expression of *cifB* in males is sometimes sufficient to cause cytoplasmic incompatibility [55–60], though paternal co-expression of both *cifA* and *cifB* genes is required for the "toxin" effect in other systems [61, 62]. Maternal expression of the *cifA* gene is sufficient to "rescue" offspring from the toxin effect [59, 60], likely by binding to CifB and thus preventing it from causing cytoplasmic incompatibility [63, 64].

Now that we've identified the genes *cifA* and *cifB*, we can introduce these two genes into the population instead of the whole bacteria to create a viable gene drive with different properties. This was previously considered in a mathematical haploid model [65]. We can also introduce the genes separately, so that *cifB* serves as a "killer" and *cifA* serves as the "rescue", thus creating a self-limiting variation of our gene drive. In this study, we present mathematical and simulation models of gene drives based on *Wolbachia* genes *cifA* and *cifB*, evaluate their basic properties and characteristics, and extend the models to continuous space to assess their performance in more realistic scenarios.

## Materials and methods

### Gene drive strategy

Our modification gene drive is based on the inheritance rules of two genes in prophage WO of *Wolbachia*: *cifA* and *cifB* (Fig 1A). Dual expression of *cifA* and *cifB* in males and no expression of *cifA* in females yields no offspring. In other words, both *cifA* and *cifB* in males are necessary to make the toxin, and only *cifA* in females is needed for an antidote. However, another study indicates that in the major malaria vector *Anopheles gambiae*, *cifB* is sufficient in males for the toxin. Therefore, we present models for both of these situations, which only produce different results in the self-limiting drive format where *cifA* and *cifB* are not part of the same allele. In our models, drive efficiency is the probability that the toxin and antidote take effect, and it is fixed to 100% unless otherwise specified. Table 1 shows all parameters in our models, including their range and default value if applicable.

### Mathematical panmictic model

We model the panmictic population model using a mathematical approach with ordinary differential equations. In our model, generations are continuous, and a generation is defined as the time it takes for a total of *K* individuals to be born and *K* individuals to die in a population with a size equal to its capacity of *K*. Note that strictly speaking, the average generation time may therefore change in some species when population size is not at equilibrium due to variance in birth or death rates, hence our definition based on equilibrium population size.

We assume that the environmental carrying capacity $K = 1$, and low density growth rate λ, which is the relative growth rate of the population when the population density is near zero, is 9. Since *K* is set to 1 we calculate the relative number of individuals of each genotype relative to carrying capacity. In our differential equations, the low density growth rate affects population growth by determining coefficients related to birth and death rates. In populations where all individuals are wild-type, to model a population that steadily increases in size to its capacity, we set the birth rate $b(N, \lambda) = \frac{N\lambda}{N(\lambda-1)+1}$ and death rate $d(N, \lambda) = N^2$. The calculation of birth

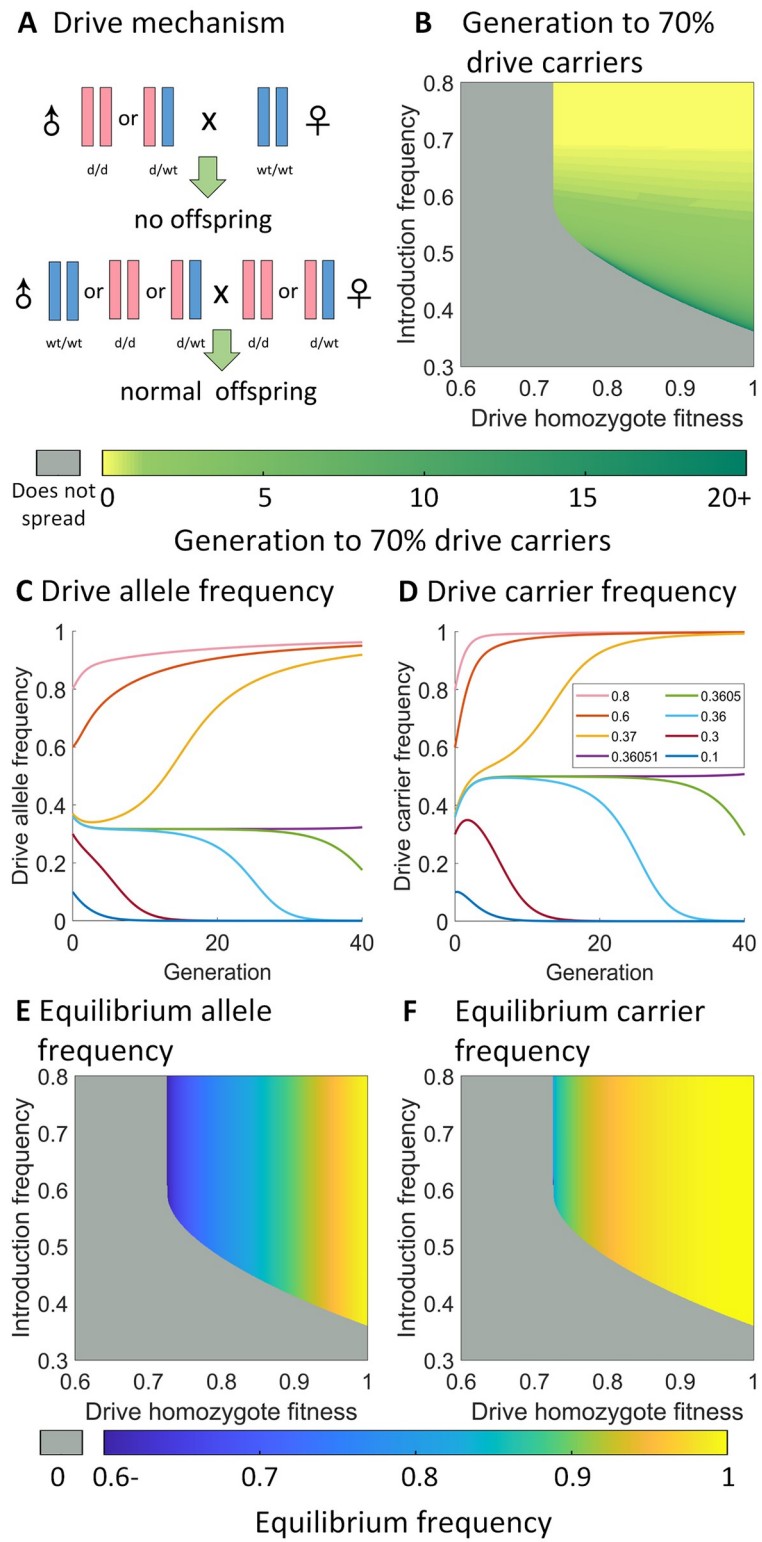

**Fig 1. Basic characterization of the CifAB drive. A**: Mechanism of CifAB drive. When male drive carriers mate with wild-type females, no viable are produced. All other possible crosses produce normal inheritance. **B**: First generation in which drive carrier frequency reaches 70%. Gray stands for inapplicable, representing cases where the drive allele is eliminated. **C,D**: Drive allele and carrier frequency trajectories after homozygote drive releases. Drive efficiency and fitness is fixed at 1. **E,F**: Drive allele and carrier frequencies at equilibrium (obtained by collecting the frequencies at

generation 1000). Gray indicates that the drive is eliminated from the population rather than reaching a high equilibrium frequency.

and death rates ensures that when $N \to 0$, $\frac{b(N,\lambda)}{N} \to \lambda$, and the relative growth rate of the population $\frac{dN}{Ndt}$ is $\lambda$. When $N = K = 1$, $b(N, \lambda) = d(N, \lambda) = 1$, and the growth rate of the population would be 0, maintaining the population size at carrying capacity. We get:

$$\frac{dN}{dt} = b(N, \lambda) - d(N, \lambda) = \frac{N\lambda}{N(\lambda - 1) + 1} - N^2$$

When a fraction of the population is drive carriers, we calculate the relative number of individuals of each genotype in our differential equations. We compute the birth rate for individuals of each genotype according to genotype fitness (representing a viability fitness cost), the relative number of individuals of different parental genotypes, and the current relative population size $N$. Similarly, the mortality rate is the relative number of individuals of the genotype multiplied by $N$, as shown below, resulting in density-dependent mortality (see below for specific equations). We note that as a population modification drive, only modest population fluctuations are expected, and these don't substantially influence the outcome in most scenarios (though slower death rates at high densities in drive release regions can improve drive performance). The density-dependent curve is fixed in all of our modeled scenarios.

For a specific genotype, (where $n$ is the relative number of individuals of this genotype, and $F$ is the genotype fitness), the population dynamics are represented as follows:

$$\frac{dn}{dt} = F \cdot \left(\sum combinations\right) \cdot \frac{\lambda}{N(\lambda - 1) + 1} - nN$$

Where $\sum$ *combinations* is calculated according to Mendel's laws of segregation and independent assortment together with drive mechanism effects. For example, in cases where *cifA* and *cifB* are in the same genetic construct, we denote the drive allele as *d* and the wild-type allele as *w*. We assume the relative number of individuals of each genotype is *dd*, *wd*, *ww*. Then, the $\sum$ *combinations* for genotype *ww* could be calculated as $ww^2 + \frac{1}{2}ww \cdot wd + \frac{1}{4}wd^2$. The $\frac{1}{2}ww \cdot wd$ term is halved because there are no viable offspring when female wild-type homozygotes mate with male drive heterozygotes.

**Table 1. Important parameters used in models of CifAB drive.**

| Symbol | Name | Definition | Value |
|--------|------|------------|-------|
| $K$ | Carrying Capacity | Wild-type population size equilibrium. | 1 |
| $N$ | Population Size | Population size relative to carrying capacity. | $= K$ at equilibrium |
| $\lambda$ | Low Density Growth Rate | Relative population growth rate when the population density is near zero. | 9 |
| $F$ | Fitness Value | Fitness of drive homozygotes (wild-type = 1). | [0.6,1], default = 1 |
| $c$ | Dominance Coefficient | Fitness dominance coefficient of the drive. | [0, 1] |
| $E_t$ | Toxin Efficiency | Probability that the toxin will take effect in offspring of male *cifA*/*cifB* carriers. | [0, 1], default = 1 |
| $E_a$ | Antidote Efficiency | Probability that the antidote will take effect in offspring of female *cifA* carriers. | [0, 1], default = 1 |
| $m$ | Migration Rate | Fraction of one population that migrate to the other population per unit time (per generation). | [0,0.1] |
| $I$ | Introduction Frequency | Initial frequency of drive individuals in the population. | [0,1) |
| $M$ | Maximum Offspring | Maximum number of offspring a female can produce in simulation models. | 50 |
| $v$ | Dispersion Factor | Standard deviation of the distance between parent and offspring in spatial simulations. | [0.01,0.1] |
| $r_m$ | Mating Radius | In spatial simulation models, females can mate with males within this distance. | $= v$ |
| $r_c$ | Competition Radius | In spatial simulation models, competition is assessed within this distance. | 0.01 |

The math model starts at $t = 0$, and in the initial population, the relative number of individuals of each genotype is determined according to the introduction frequency. The total size of the initial population is 1.

To model the fitness costs that drive alleles may impose on their carriers, we refer to the relative fitness of drive homozygotes compared to wild-type homozygotes as $F$. Normally, we assume that fitness costs are multiplicative, so that the fitness for drive/wild-type heterozygotes is $\sqrt{F}$. However, when we assess the impact of drive allele dominance coefficient on gene drive performance, we calculate the genotype fitness in a different way. When the dominance coefficient $c \in [0, 1]$, the fitness for heterozygotes is $1 - c(1 - F)$.

Drive component efficiency may vary according to factors like climate and insect age. It may have a substantial impact on drive performance [66, 67]. We consider drive component efficiency as well for models where *cifA* and *cifB* are in the same genetic construct. We calculate the efficiencies of both the toxin and antidote. Toxin efficiency $E_t$ is the probability that the toxin will take effect in offspring of male *cifA*/*cifB* carriers, and antidote efficiency $E_a$ is the probability that the antidote will take effect in offspring of female *cifA* carriers. In our model, we assume that efficiency reductions are multiplicative, and both $E_t$ and $E_a$ stand for the efficiency value per allele. For example, male drive homozygotes have a toxin efficiency of $1 - (1 - E_t)^2$, and male drive/wild-type heterozygotes have a toxin efficiency of $E_t$. Then, we can calculate the chance that the toxin takes effect in the progeny of drive males and the antidote doesn't take effect in the progeny of drive females, determining the offspring mortality rate for each combination of parental genotypes. For example, the chance that male drive homozygotes and female drive heterozygotes cannot produce offspring is $[1 - (1 - E_t)^2] \cdot (1 - E_a)$.

To evaluate the effects of migration between two linked panmictic subpopulations on the spread of this gene drive, we refer to the migration rate, which is the fraction of individuals in each subpopulation that migrate to the other subpopulation per unit of time, as $m$. Then, based on calculations of birth and death rates of the specific genotype in both subpopulations, which we denote as $b_1$, $d_1$ and $b_2$, $d_2$, respectively, for subpopulations one and two, we can derive the differential equations for this genotype in both subpopulations. $n_1$ and $n_2$ refer to the numbers of individuals relative to carrying capacity of a given genotype in subpopulations one and two, and $g_1$ and $g_2$ refer to the growth rate of this genotype in populations one and two. Specifically, $g_1 = b_1 - d_1$, $g_2 = b_2 - d_2$. we derive the following differential equations:

$$\frac{dn_1}{dt} = (1 - m)g_1 + mg_2$$

$$\frac{dn_2}{dt} = (1 - m)g_2 + mg_1$$

In our model, we vary several parameters including introduction frequency, drive efficiency, drive fitness, fitness dominance coefficient, and migration rate. When *cifA* and *cifB* are in the same genetic construct, the overall ability of drive carriers to increase in frequency in the population depends on the drive carrier frequency because there is no offspring when males have the drive allele and females don't have the drive allele. Therefore, the drive introduction frequency has to be above a certain level for the drive allele to eventually be fixed in the population or increase to its high equilibrium frequency. To collect the threshold, we solve the differential equations numerically with the MATLAB function ode45 and search for the minimum introduction frequency required for the drive carrier frequency to exceed 70% at generation 1000, when genotype frequencies are sure to have reached an equilibrium (this equilibrium frequency is always above 70% when a nonzero equilibrium is possible). We use bisect search to find the threshold for each set of parameters. We calculate the average drive

allele and carrier frequencies in the first 100 generations by dividing the integral of the absolute number of drive alleles or drive carriers by the integral of population size. For example, assume $dd(t)$, $dw(t)$, $ww(t)$ are the number of individuals of drive homozygotes, heterozygotes, and wild-type homozygotes at time $t$. The average drive allele frequency the first 100 generations after drive release is $\frac{\int_0^{100}\left(dd(t)+\frac{1}{2}dw(t)\right)dt}{\int_0^{100}(ww(t)+dw(t)+dd(t))dt}$, and the average carrier frequency is

$\frac{\int_0^{100}(dd(t)+dw(t))dt}{\int_0^{100}(ww(t)+dw(t)+dd(t))dt}$. We also collect the first generation in which drive carrier frequency exceeds 70% if applicable.

For self-limiting variations of the drive where *cifA* and *cifB* are in different genetic constructs, we also collect the protection time for each set of parameters, which is the total time that *cifA* carrier frequency is above 80% (any cargo effector gene is assumed to be with the *cifA* allele). Since self-limiting drives are always bound to be lost if *cifA* has a fitness cost, protection time is a good indicator of the effectiveness of the drive. We also record the maximum drive allele and carrier frequency and the generation in which it reaches its maximum.

### Equations for the 1-deme panmictic model

Let $x_1$, $x_2$, $x_3$ be the number of individuals of genotypes: drive homozygotes (*dd*), drive/wild-type heterozygotes (*dw*), wild-type homozygotes (*ww*), relative to carrying capacity $K$. $F$ is the relative fitness of drive homozygotes. $E_t$, $E_a$ stands for toxin efficiency and antidote efficiency. We introduce drive homozygotes at introduction frequency $I$ during initialization when $t = 0$.

At initialization, $x_1 = I$, $x_2 = 0$, $x_3 = 1 - I$. $\lambda$ is the low-density growth rate. The fitness for genotype *dd* is $F_{dd} = F$. When we assume that fitness costs are multiplicative, the relative fitness value for genotype *dw* is $F_{dw} = \sqrt{F}$. When we calculate fitness cost according to dominance coefficient $c$, $F_{dw} = 1 - c(1 - F)$. Then, we use the following ordinary differential equations to calculate the relative number of individuals of each genotype.

$$N = x_1 + x_2 + x_3.$$

$$\frac{dx_1}{dt} = F_{dd}\left(\sum combinations(dd)\right)\frac{\lambda}{(\lambda - 1)N + 1} - x_1 N$$

$$\frac{dx_2}{dt} = F_{dw}\left(\sum combinations(dw)\right)\frac{\lambda}{(\lambda - 1)N + 1} - x_2 N$$

$$\frac{dx_3}{dt} = \left(\sum combinations(ww)\right)\frac{\lambda}{(\lambda - 1)N + 1} - x_3 N$$

In which

$$\sum combinations(dd) = x_1^2\left(1 - \left(1 - (1 - E_t)^2\right)(1 - E_a)^2\right) + \frac{1}{4}x_2^2(1 - E_t(1 - E_a)) +$$

$$\frac{1}{2}x_1 x_2\left(2 - \left(1 - (1 - E_t)^2\right)(1 - E_a) - E_t(1 - E_a)^2\right)$$

$$\sum combinations(wd) = \frac{1}{2}x_2^2(1 - E_t(1 - E_a)) + \frac{1}{2}x_2 x_3(2 - E_t) + x_1 x_3\left(2 - \left(1 - (1 - E_t)^2\right)\right) +$$

$$\frac{1}{2}x_1 x_2\left(2 - E_t(1 - E_a)^2 - \left(1 - (1 - E_t)^2\right)(1 - E_a)\right)$$

$$\sum combinations(ww) = x_3^2 + \frac{1}{2}x_2 x_3(2 - E_t) + \frac{1}{4}x_2^2(1 - E_t(1 - E_a))$$

After calculating the relative number of individuals of each genotype, we calculate the genotype frequencies for each point in time, thus calculating allele and carrier frequencies. If the relative number of individuals of genotype dd, dw, ww are $x_1$, $x_2$, $x_3$ at a specific time point, the drive allele frequency is $\frac{x_1+\frac{1}{2}x_2}{x_1+x_2+x_3}$, and the drive carrier frequency is $\frac{x_1+x_2}{x_1+x_2+x_3}$.

## Mathematical spatial model

To evaluate the performance of the drive in potentially more realistic continuous space scenarios, we extend our mathematical model into 2D space by using partial differential equations. Our calculations are done on a 1x1 arena. Similar to the panmictic mathematical model, the $N(x, y)$ in the equations stands for the total of all individual genotypes, but in this model, instead of the absolute value of population size, it can be understood as the local population density relative to carrying density at point $(x, y)$ in the arena. According to our population growth model, carrying density, which is the capacity of population density at each point, is also $K$. Thus, we can reuse the calculation for growth and death rates in our panmictic model. Our equations are shown below, where $n(x, y)$ represents the absolute density of a specific genotype at point $(x, y)$. $b(N, \lambda)$ and $d(N, \lambda)$, respectively, stand for the birth and death rates of the specific genotype at point $(x, y)$, which are calculated in the same way as the panmictic model:

$$\frac{\partial n}{\partial t} = D \cdot \frac{\partial^2 n}{\partial x^2} + D \cdot \frac{\partial^2 n}{\partial y^2} + b(N, \lambda) - d(N, \lambda)$$

In the equation, $D = \frac{1}{2}v^2$ and reflects the movement rate of individuals. In our spatial model, we keep *cifA* and *cifB* in the same genetic construct and assume that fitness costs are multiplicative.

The drive can potentially travel in a wave through the population in continuous space. To measure wave speed, we add drives in the left 30% of the arena ($0 \leq x \leq 0.3$). In this scenario, the status of all points with the same x-coordinate is the same. Therefore, we can take a slice through the arena parallel with the x-direction and simplify the 2-dimensional partial equations to 1D:

$$\frac{\partial n}{\partial t} = D \cdot \frac{\partial^2 n}{\partial x^2} + b(N, \lambda) - d(N, \lambda)$$

We initialize the simulation by filling the left 30% with drive individuals, and the rest is filled with wild-type individuals. The density at all positions is equal to carrying density. We then choose two points to calculate wave speed. One is at $x_1 = 0.5$, and the other is $x_2 = 0.7$. We detect the number it takes for drive carrier frequency to reach 50% for the first time in each of the two detection points. Assuming that it takes $t_1$ generations to reach $x_1$ and $t_2$ generations to reach $x_2$, we can calculate the drive wave speed using the equation $\frac{x_2-x_1}{t_2-t_1}$.

We also model radially symmetric drive releases, where drive individuals are placed in a circle at the center of the arena. At initialization, the density of wild-type individuals is 1, which is equal to carrying density, all over the arena. Inside the drive release area, extra drive homozygotes are added to the population, which will increase the density above the carrying density. We define release density as a function of the distance to the center of the arena. In circular releases, all points that have the same distance to the center of the arena share the same status. Therefore, we can make a transformation from rectangular to polar coordinates, taking advantage of radial symmetry. In this transformation, by using $x = r cos\theta$, $y = r sin\theta$, we can get $\frac{\partial^2 n}{\partial x^2} + \frac{\partial^2 n}{\partial y^2} = \frac{\partial^2 n}{\partial r^2} + \frac{1}{r^2}\frac{\partial^2 n}{\partial \theta^2} + \frac{1}{r}\frac{\partial n}{\partial r}$. To match the square arena with an edge length of 1, we set the

maximum radius to 0.5. Based on the radial symmetry of the scenario, $\frac{\partial^2 n}{\partial \theta^2} = 0$. So we can simplify this scenario to 1D where radius is the only spatial parameter.

$$\frac{\partial n}{\partial t} = D \cdot \left( \frac{\partial^2 n}{\partial r^2} + \frac{1}{r} \frac{\partial n}{\partial r} \right) + b(N, \lambda) - d(N, \lambda)$$

Since our calculations are done in 1D, we have to integrate the population density over space to calculate the population size and drive frequencies of the whole population. Assuming that the genotype relative densities for wild-type homozygotes, heterozygotes, and drive homozygotes are $ww(r)$, $wd(r)$ and $dd(r)$, we can calculate the carrier frequency of the whole circular arena by using $\frac{\int_0^{0.5} 2\pi r(wd(r)+dd(r))dr}{\int_0^{0.5} 2\pi r(ww(r)+wd(r)+dd(r))dr}$. Using this method, we calculate the drive carrier frequency at generations 10 and 15, and compare them to see if the drive frequency has increased, thus determining if the drive circle is still expanding after the dispersal effects in the first few generations.

To investigate the optimal drop shape of radially symmetric drops, we model a ring drop and a radial linear drop. Similar to the panmictic population model, there are also threshold-like properties in the spatial model. When the total drop size is below a certain value, the drive allele and carrier frequencies are bound to shrink over time, and the carrier frequency in generation 15 will be less than that of generation 10. Therefore, we're interested in the critical drop size, which is the minimum drop size required for the drive to expand in generations 10 through 15, relative to carrying capacity. We're also interested in the exact shape of the drop, or in other words, the function $dd(r)$ at initialization, when the expansion drop size is at its minimum.

In the radial linear drop, we assume that the function $dd(r)$ is linear inside a certain radius range, and when the distance of a certain point to the center of the arena is $r$, the relative density of drive homozygotes at that point can be expressed as $kr + b$, where $k$ and $b$ are constants. With a larger drop radius, the drive is more likely to persist and expand, and we can define the critical radius as the minimum drop radius that allows the drive to expand instead of contract. Fixing the drive homozygote fitness value $F$ to 1, we can collect the critical radius $R$ for each $k$ and $b$ value. The critical radius is the minimum drop radius required, for the drop to keep expanding after generation 10. In this case, $dd(r) = \begin{cases} kr + b & , r \in [0, R] \\ 0 & , r \in (R, 0.5] \end{cases}$. Then, we calculate the critical relative drop size $V_c$ by integrating the function $dd(r)$ over the arena, which is $\frac{2}{3}\pi k R^3 + \pi b R^2$. The uniform circle drop is a special situation of a radically linear drop, in which $k = 0$, and $b = \frac{I}{1-I} = 4$ since we fix introduction frequency $I$ at 0.8. In this case, we can assess the effects of the drive fitness, as well as dispersion factor. The critical drop size $V_c = \pi R^2 \frac{I}{1-I}$.

In the ring drop, drive homozygotes are dropped uniformly in a ring. Let drive homozygote relative density in the ring be $h$, assume that the inner radius of the ring is $r_0$ and the thickness of the ring is $d$. In this case after initialization, drive homozygote genotype relative density is set at $dd(r) = \begin{cases} h & , r \in [r_0, r_0 + d] \\ 0 & , r \in [0, r_0) \cup (r_0 + d, 0.5] \end{cases}$. For each set of $r_0$ and $d$, we collect a critical drive density $h$ and calculate the critical relative drop size $V_c = h\pi(2r_0 d + d^2)$. We assume that $h \leq 99$, since it's unrealistic to have an introduction frequency of over 99%, and we consider the situation to be not applicable if the drive is unable to keep expanding at any $h$ that is below 99.

## Equations for the mathematical spatial model

In the spatial model, we only consider the scenario where *cifA* and *cifB* are in the same genetic construct, and the symbols $\lambda$, $N$, $F$, $E_t$, $E_a$ have the same definition as the panmictic model. $\sum combinations(dd)$, $\sum combinations(dd)$, $\sum combinations(dd)$, $F_{dd}$, $F_{dw}$, as well as allele and carrier frequencies, are also calculated using the equations above. The only difference is that $x_i (i = 1, 2, 3)$ is a function of not only time $t$ as in the panmictic model but also of $x$, $y$ in the spatial model. Therefore in this model, $x_i$ represents the population density at a specific point in space and time relative to carrying density. $D$ can be calculated from the dispersion factor $v$ by using $D = \frac{1}{2} v^2$.

Our partial differential equations are as follows:

$$\frac{\partial x_1}{\partial t} = D\left(\frac{\partial^2 x_1}{\partial x^2} + \frac{\partial^2 x_1}{\partial y^2}\right) + F_{dd}\left(\sum combinations(dd)\right)\frac{\lambda}{(\lambda - 1)N + 1} - x_1 N$$

$$\frac{\partial x_2}{\partial t} = D\left(\frac{\partial^2 x_2}{\partial x^2} + \frac{\partial^2 x_2}{\partial y^2}\right) + F_{dw}\left(\sum combinations(dw)\right)\frac{\lambda}{(\lambda - 1)N + 1} - x_2 N$$

$$\frac{\partial x_3}{\partial t} = D\left(\frac{\partial^2 x_3}{\partial x^2} + \frac{\partial^2 x_3}{\partial y^2}\right) + \left(\sum combinations(ww)\right)\frac{\lambda}{(\lambda - 1)N + 1} - x_3 N$$

## Simulation panmictic model

To evaluate the gene drive from an individual-based point of view and model stochasticity that can be found in real-world scenarios, we used a simulation model of a panmictic population with SLiM software [68] that is similar to our previous studies [69, 70].

In our simulation panmictic model, we focus on a population of sexually reproducing diploids with non-overlapping generations where each individual has specific sex and genotype. Fitness is calculated according to the genotype of the individual. We assume fitness costs only have an impact on female fecundity and male mating success in the reproduction phase.

We use our population growth model to adjust the population size. We assume that in the simulation model, when the population density is very low and all individuals are wild-type, the ratio of population size in two successive generations is $\frac{N_{i+1}}{N_i} = \lambda + 1$. We set $\lambda$ to 9 as in our mathematical model so that the population can experience a 10-fold increase in size per generation at a very low density.

In each generation, the individuals of the population mate and produce offspring. Each female randomly selects a male in the population, and the male is chosen as her mate at the probability equal to his relative fitness value. Otherwise, the female randomly selects another candidate. If the female rejects 10 candidates in a row, she will not reproduce. If the pairing is fertile despite possible toxin effects, the expectation of the number of offspring she produces is equal to twice her fitness value multiplied by a density-dependent scaling factor $\sigma = \frac{\lambda+1}{\lambda\frac{N}{K}+1}$, so that each wild-type female may have an average of 2 offspring when the population size reaches its capacity, and about $2(\lambda + 1)$ offspring when the population size is very low. We assume that females can have a maximum of $M = 50$ offspring. The actual number of offspring the female produces is drawn from a binomial distribution, with $M$ trials and $p = 2F\sigma/M$, where $F$ is the female's genotype fitness value. The sex of each offspring is random, and its genotype is determined by randomly selecting one allele from each parent for each locus.

In the simulation model, we calculate the drive allele and carrier frequency after each generation. If the drive carrier frequency can reach 99% (which is below the equilibrium of 100% since the drive has no fitness costs in this scenario) in the first 300 generations, we consider the drive to have been established in the population. Otherwise, if the drive allele is eliminated

from the population or doesn't fix before generation 300, we consider it as an unsuccessful release. We repeat the simulation 200 times and calculate the drive establishment rate.

## Simulation spatial model

We also extend our simulation model into 2D space. As in the panmictic model, it's an individual-based model with discrete, non-overlapping generations, with the same reproduction rules. All individuals are located in a 1x1 (unitless) arena. Though we use SLiM software for both simulation models, there are some slight differences in the spatial model compared to the panmictic model.

Instead of randomly choosing a male as a candidate for mating, each female can only choose from the male candidates inside the mating radius $r_m$. In the spatial model, population size is regulated by local population density instead of total population size. To calculate the density-dependent scaling factor $\sigma$ in this model, we consider the competition of individuals in the radius of $r_c$, the competition radius. The competition intensity between individuals is a function of their distance $r$, $i(r) = 1 - \frac{r}{r_c}$. We can calculate the expected competition sum $c_e = N \int_0^{r_c} 2\pi r \cdot i(r) dr$. Then, for each individual in space, we calculate the actual competition sum $c_a = \sum i(r)$ for all other individuals inside the radius of $r_c$ near the individual. Instead of $\frac{\lambda+1}{\lambda\frac{N}{K}+1}$, the density-dependent scaling factor $\sigma$ in this model is calculated as $\frac{\lambda+1}{\lambda\frac{c_a}{c_e}+1}$. When females reproduce, the number of offspring is calculated in the same way as the panmictic model, using the new $\sigma$.

After an offspring is produced, it is displaced from the mother to avoid clustering and represent the distance an individual travels from birth to reproduction. The displacement in both x and y directions is drawn from a normal distribution, with a mean of 0 and standard deviation equal to the individual dispersion factor $v$. This means that the radius of the displacement will have a standard deviation of $v$ in a random direction, and an average of $\sqrt{\frac{\pi}{2}} \cdot v$. The position of offspring outside the boundaries is regenerated until they fall within the boundaries. We can convert the different terms of dispersion in the mathematical and simulation models using the formula $D = \frac{1}{2}v^2$, thus matching the results of both models and enabling comparison.

In the spatial simulation model, we measure wave speed with a method similar to the measurement in the mathematical model. We fill the left $30\%(0 \leq x \leq 0.3)$ of the arena with drives, and extract two strips of the arena which are 0.05 in width and 1 in length. They are placed in $0.5 \leq x \leq 0.55$ and $0.7 \leq x \leq 0.75$. We calculate the drive carrier frequency in both strips in every generation, and record the two generations in which one of them first exceeds 50%, denoted as $t_1$ and $t_2$. The distance between the centers of the two strips is $d_s = 0.2$, and therefore, we calculate the wave speed as $\frac{d_s}{t_2-t_1}$.

## Data collection

We collected data for both the mathematical models and simulation models using the High-Performance Computing Platform of the Center for Life Science at Peking University. We used python to process data and MATLAB to prepare figures. All MATLAB and SLiM models are available at https://github.com/jchamper/ChamperLab/tree/main/CifAB-Drive-Modeling.

## Results

### Characterization of CifAB drive in panmictic populations

First, we model a gene drive in which *cifA* and *cifB* are in the same genetic construct and are introduced at one locus, which we call the CifAB drive. If a male drive carrier mates with a

wild-type female, they will have no offspring due to cytoplasmic incompatibility (Fig 1A). As long as the drive carrier frequency is high enough, the drive can spread in the population even if it has a fitness cost, since when mating with male drive carriers, female drive carriers can have viable offspring while female wild-type individuals cannot. Female drive carriers are at an advantage compared to wild-type if the frequency of male drive carriers is high enough. In contrast, male drive carriers have a lower chance of successful reproduction compared to male wild-type individuals, especially if the drive carrier frequency is low, thus reducing the frequency of the drive in the next generation due to removal of paternal drive alleles. Therefore, there exists an introduction threshold for this type of drive, below which the drive will be eliminated. This is not the case for *Wolbachia* bacteria, which lack an introduction threshold in ideal form and only gain a threshold if there are fitness costs or imperfect transmission efficiency [71–73]. This is because *Wolbachia* bacteria are maternally transmitted, so no *Wolbachia* transmission is lost when infected males fail to produce offspring with wild-type females. In this sense, CifAB drive's population dynamics are perhaps more similar to underdominance type gene drive systems. Below, we explore some basic properties of the CifAB drive.

**Basic performance characteristics.** The 1-deme panmictic release scenario models a panmictic population with a capacity of 1 in the mathematical model. The scenario was initialized by introducing wild-type and drive individuals. The fraction of drive homozygous individuals initially present in the population was the introduction frequency $I$, and the total number of individuals was equal to carrying capacity $K$.

We varied the drive homozygote fitness $F$ of the CifAB drive allele relative to wild-type allele from 0.6 to 1. We assumed that the fitness effects were multiplicative, so the fitness of drive heterozygotes would be $\sqrt{F}$. For each fitness value, we used our math model to collect the threshold, which is the minimum introduction frequency needed for the drive to fix in the population, or reach a high frequency at equilibrium. (S1(A) Fig) When $F < 0.72$, the threshold is 1, which indicates that the drive allele is always lost as long as there were any wild-type individuals initially in the population. We also collected the generation when drive carrier frequency first reached 70% and found that in most cases when drive equilibrium frequency ≥ 70%, carrier frequency was able to reach 70% in the first 20 generations (Fig 1B), indicating that the drive can spread rapidly in a panmictic population. Example allele and carrier frequency trajectories with various release sizes clearly show a threshold of slightly below 37% (Fig 1C and 1D), with the drive frequency eventually going to 100% if released above the threshold. In cases where one drive allele is sufficient for the drive's effector to be efficient at its task (such as elimination of disease transmission), the equilibrium carrier frequency is a good indicator of the drive's long-term performance. The CifAB drive equilibrium carrier frequency was observed to approach 1 substantially more quickly than the allele frequency with increasing fitness (Fig 1E and 1F).

The introduction frequency threshold in the absence of fitness costs is 36.051% at a precision of 0.001% (somewhat higher than a haploid model [65] due to the initial population consisting of homozygotes rather than a mix near equilibrium). This is below 50% because wild-type females and an equal number of male drive carriers who they mate with are the only individuals that will fail to reproduce. Wild-type females will fail to pass on two wild-type alleles, and male drive homozygotes will fail to pass on two drive alleles. However, male drive heterozygotes will fail to pass on one wild-type and one drive allele, thus creating a situation in which more wild-type alleles are removed from the population than drive alleles when drive heterozygotes are present, giving the drive an overall advantage when there are even numbers of drive and wild-type alleles in the population. In our carrier frequency trajectory (Fig 1D), we can see that in cases where introduction frequency is 36.05% and 36.051% (slightly below and above

the threshold, respectively), carrier frequency quickly reaches a value near 50% for both situations. Afterwards, the carrier frequency trajectory lines depart and the drive is either fixed or lost. Therefore we can conclude that 50% carriers (only after a balance is reached between drive heterozygotes and homozygotes) is an unstable equilibrium for CifAB drives with zero fitness cost. From a biological point of view, this is because when drive carrier frequency is exactly 50%, the reproduction disadvantage of male drive carriers and female wild-type individuals is the same. We can assume that for each genotype, the sex ratio is 1:1 and define reproduction success rate as the probability an individual can successfully mate and produce viable offspring. If we denote wild-type frequency as $x_1$ and drive carrier frequency as $x_2$, the reproduction success rate for drive carriers would be $\frac{1}{2}\left(\frac{x_2}{x_1+x_2} + 1\right)$ and the reproduction success rate for wild-type individuals would be $\frac{1}{2}\left(\frac{x_1}{x_1+x_2} + 1\right)$. When $x_1 = x_2$, the two success rates are equal. When drive carrier frequency > 50%, $x_1 < x_2$ and drive carriers have a larger reproduction success rate than wild-type individuals, resulting in further spread of the drive allele.

To further investigate the properties of the drive, we varied introduction frequency $I$ from 0.3 to 0.8, while varying drive homozygote fitness $F$ from 0.6 to 1, and collected the average drive allele and carrier frequencies in the first 100 generations after the drives were dropped (S1B and S1C Fig). As expected, both average allele and carrier frequencies increased with fitness and introduction frequency. In many cases, the carrier frequency reached 1 when the allele frequency was lower. We also collected the equilibrium allele and carrier frequencies (Fig 1E and 1F). In cases with a relatively large fitness cost ($0.72 \leq F \leq 0.76$), the equilibrium allele and carrier frequencies are all less than 1. They are only influenced by fitness, with introduction frequency simply determining if the drive will eventually reach these levels or be eliminated.

Drive efficiency is the probability that drive alleles can be adequately expressed in the correct time, location, and quantity to successfully invoke reproduction manipulations. Specifically, this means the drive allele in males will cause the toxin effect in offspring, and drive allele in females will rescue offspring from the toxin effect.

In our model, $E_t$ and $E_a$, respectively, denote the male toxin efficiency and female antidote efficiency. In previous results, we fix both $E_t$ and $E_a$ to 1, but to assess the impact of drive efficiency, we hold fitness at 1 and vary $E_t$ and $E_a$ from 0.2 to 1. Our results show that the threshold of the drive mainly depends on antidote efficiency, decreasing with higher antidote efficiency (Fig 2A). Toxin efficiency only has a slight effect, with higher toxin efficiency usually decreasing the threshold, though the pattern is weakly reversed when antidote efficiency is high. Specifically, when antidote efficiency is 1, the threshold decreases from 36.05% to 34.39% as toxin efficiency is reduced from 1 to 0.2. When toxin and antidote efficiencies are changed together, we find that the introduction threshold decreases with the increase of drive efficiency, and the drive is lost for any drive introduction frequency below 1 when drive efficiency is lower than 0.36 (S2(A) Fig).

Fitness effects can be complicated, particularly if any cargo gene affects other biological pathways in the target organism. To further investigate the effect of fitness on drive performance characteristics, we independently vary the dominance coefficient $c$ in addition to the homozygote fitness value. In this case, we no longer assume that the fitness cost is multiplicative. Instead, the genotype fitness for drive/wild-type heterozygotes is calculated as $1 - c(1 - F)$. We keep drive efficiency at 1 and collect the introduction threshold (Fig 2B) and the average drive allele and carrier frequencies in the first 100 generations (S2B and S2C Fig). When the fitness dominance is varied, the threshold is always above 0.36, which indicates that the drive retains its confined properties. The fitness value has a much larger effect on the value of

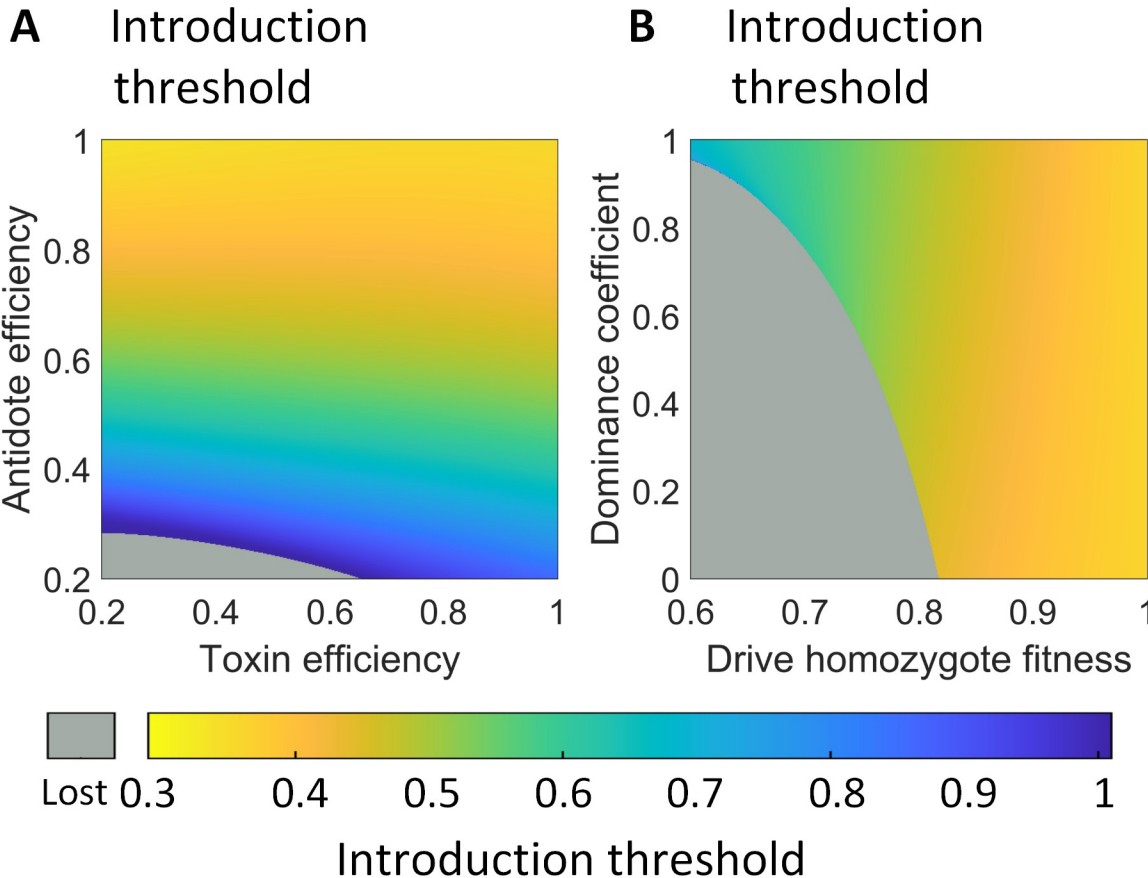

**Fig 2. Effect of drive performance parameters on the introduction threshold. A**: The introduction threshold as a function of toxin and antidote efficiency. **B**: The introduction threshold as a function of drive fitness value and its dominance coefficient. Gray indicates that the drive is lost for any drive starting frequency below 1.

the threshold than the dominance coefficient, though a higher dominance coefficient does slightly increase the threshold. Additionally, when the dominance coefficient is low, the introduction threshold tends to skip from an intermediate value directly to a parameter regime where no introduction frequency will allow the success of the drive. This occurs at the point of convergence between the drive's introduction frequency threshold and its equilibrium frequency.

**Confinement in a 2-deme scenario.** A 2-deme scenario is one of the most common for assessing drive confinement. In this scenario, we model two panmictic populations linked by reciprocal migration. We add drive individuals to the introduction deme and vary both introduction frequency and migration rate. We fix the drive homozygote fitness value to 0.95 (representing an efficient but imperfect drive) and collect the average allele and carrier frequencies in the first 100 generations, as well as the first generation when drive carrier frequency reaches 70% (S3 Fig), for both demes.

When the migration rate *m* is low (below 0.06), the drive carrier frequency in the introduction deme is more likely to rapidly increase, and when the introduction frequency is high enough for establishment, drive carrier frequency may even reach a high frequency in the introduction deme (Fig 3). The drive frequency in the linked deme will be maintained at a low frequency because the migration between the two demes is too low to allow it to establish in

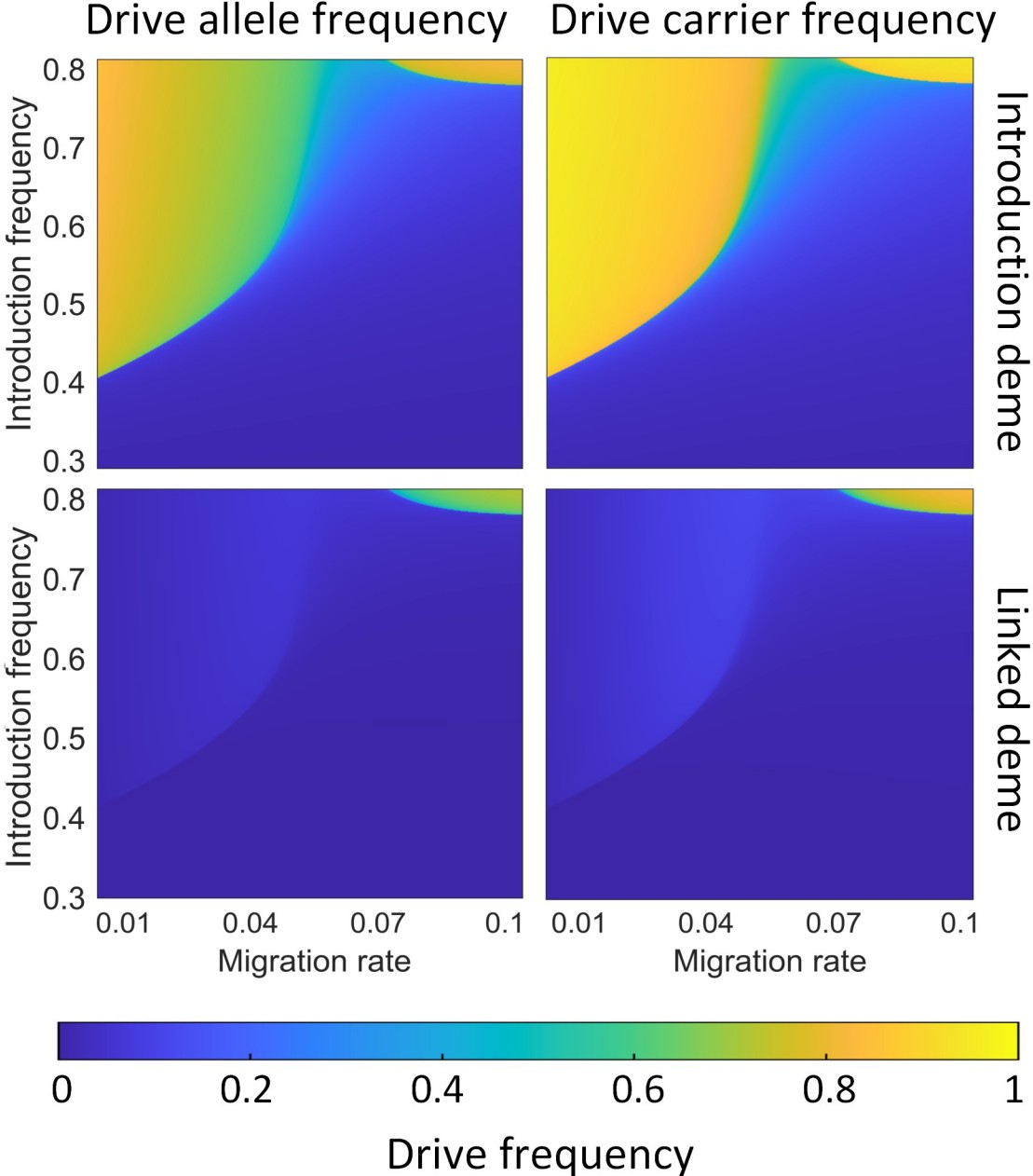

**Fig 3. Confinement of the CifAB drive in a scenario with two linked demes.** We show the average allele and carrier frequencies of the CifAB drive in the first 100 generations after adding drive individuals in both the introduction deme and the linked deme. Blue colors represent areas of the parameter space where the drive is lost or where it fails to invade the second deme and is present only at a low equilibrium frequency.

the linked deme. When migration and introduction rates are both high enough, the drive allele may successfully invade the linked deme. However, when $0.06 \leq m \leq 0.075$, it's difficult for the drive to establish in either deme, since the migration rate is enough to weaken the drive's effect on the introduction deme (due to the influx of wild-type individuals), yet not high enough to ensure that enough individuals migrate to the linked deme to enable establishment there.

A similar scenario involves a single deme with a continuous influx of drive individuals, presumably from a separate deme that does not receive any migration from the focal deme. In this case, a drive homozygote influx of approximately 4.5% of the total population per generation will be sufficient to eventually allow the drive to fully spread through the deme. At lower levels of migration, a low equilibrium will be reached with the drive allele frequency staying below 15%.

### Drive performance in continuous space

**Drive wave speed.** Next, we extend our drive model to continuous space and examine its spatial properties. We initialize the model by setting the left 30% of the arena as all drive individuals, and the right 70% all wild-type. As time elapses, the drive can form a wave of advance as more individuals become drive carriers. The drive wave speed is the average speed the drive spreads across the arena. When $0.01 \leq v \leq 0.1$, we collect the drive wave speed in both mathematical and simulation models and compare the results (S4(A) Fig).

Using the mathematical model, we varied both the individual dispersion factor $v$ and drive homozygote fitness $F$, and we collected the wave speed, which is the average distance relative to the length of the arena that the drive wave could travel in each generation (Fig 4A). The wave cannot advance at all when the fitness value is below 0.92 because the drive has no advantage compared to the wild-type allele. A drive wave of advance cannot build up, and the front line of the drive recedes as wild-type takes over the population. When the wave is able to

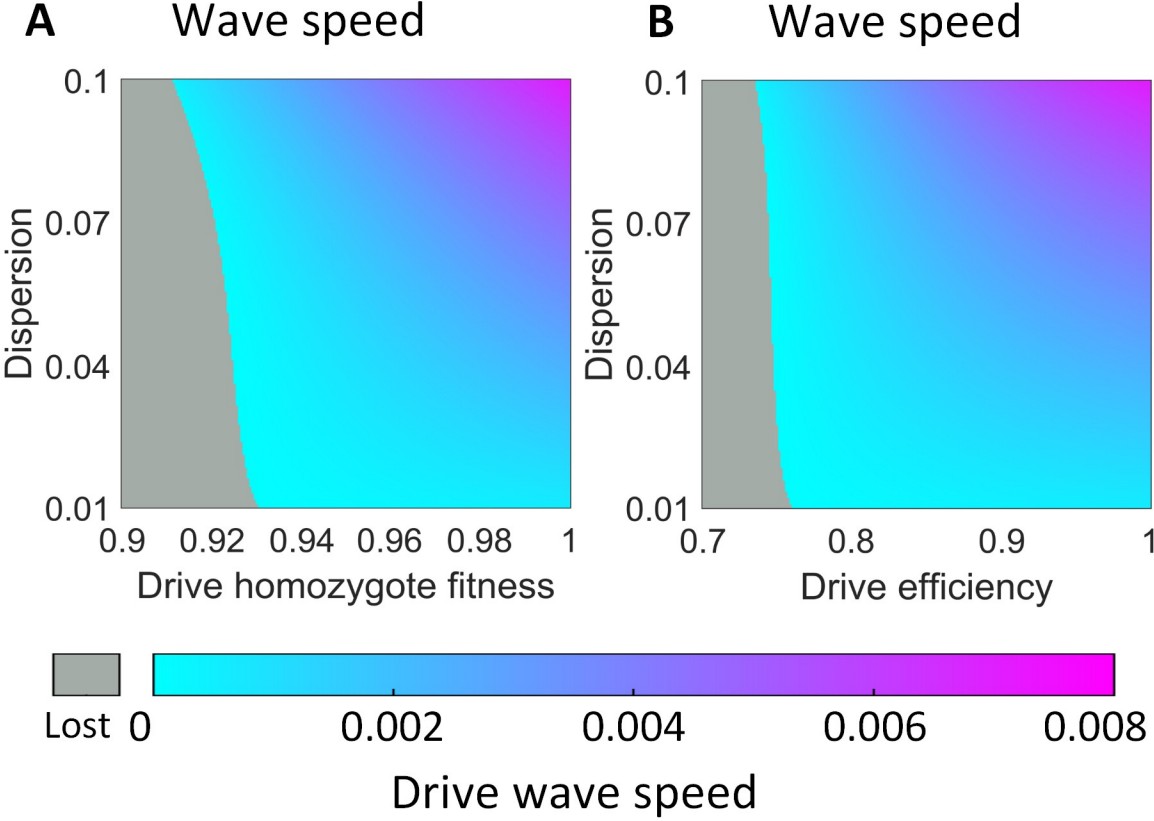

**Fig 4. Drive wave speed.** The drive wave advance speed is displayed (in unitless distance units per generation) as a function of dispersion and either **A**: drive fitness or **B**: drive efficiency. Gray indicates that a drive wave is not able to form.

establish, the wave speed is proportional to the individual dispersion rate since a higher dispersion enables drive carriers to spread faster across the arena.

We also assessed the effect of drive efficiency on wave speed for a drive without fitness costs (Fig 4B). We assumed that toxin and antidote efficiencies were equal and varied both efficiencies together as well as varying dispersion factor *v*. We found that drive efficiency plays a critical role in wave formation, and the wave is unable to form when drive efficiency is below 0.74. When the drive wave is able to form, wave speed increases along with drive efficiency and dispersion.

**Spatial radial release scenario.**　If a gene drive is capable of spreading, a single, central release in a target area would often be desirable to minimize the resources required for the release while still enabling the drive to eventually reach the whole population. Previous studies of confined drives and underdominance alleles have shown that the critical release amount (referring to the minimum number of released drive individuals needed for success) must be sufficient for the drive to spread, making it particularly important to consider for this type of drive [33]. In our spatial circle release scenario, after releasing drive individuals, the population density around the center will exceed the carrying density. Some of the extra individuals will die due to increased competition, and others will disperse out of the drop circle. Due to this natural dispersion, the drive circle will likely expand in the first few generations before the average population density falls to a normal level. After that, the drive may keep expanding, or the drive area may contract. If it contracts, the total drive carrier frequency will decrease, and the drive will eventually be lost. In our study, we examine different release patterns, searching for optimal parameters to ensure that the drive will keep expanding after releasing the minimum number of drive individuals.

The uniform circle release involves an even release of drive individuals throughout a circle at a level above their introduction threshold (Fig 5A). This was a type of release pattern investigated previously for toxin-antidote drives in continuous space [33]. We set the introduction frequency to 0.8. We then use this scenario to assess the effects of drive homozygote fitness and dispersion on drive performance. We find that drive performance decays rapidly with even a mild relative fitness of 0.95, as expected due to the drive's inability to form a wave of advance at lower fitness values in even neutral situations. Furthermore, the critical drive homozygote release size is lower at low dispersion values. This is because when the dispersion is relatively high, more drive individuals disperse to areas with low drive density, where they may be eliminated from the local population since the drive frequency is not high enough for the drive allele to spread. Also, because the drive is surrounded by wild-type, higher dispersion tends to bring in more wild-type individuals from surrounding areas, reducing the local drive frequency below its introduction threshold and causing the drive area to shrink.

In the linear radial drop (Fig 5B), when the radius is *r*, the relative drive density $dd(r)$ is a linear function of radius when $0 \leq r \leq R$, where *R* is the critical radius. Assuming that the drive has no fitness cost and with a dispersion factor *v* = 0.03, we can express the linear function using $dd(r) = kr + b$, $0 \leq r \leq R$, and collect the critical radius for each set of *k* and *b*, thus determining the critical release size $V_c$. This allows the drive release pattern to have a lower central density, potentially increasing with a higher radius. The optimal $V_c$ for linear radial release is 0.245, when *k* = 5.43, *b* = 0.03, *R* = 0.28. For the more commonly modeled uniform circle release where *k* = 0, the critical release size is rather higher at 0.33. This is because when *b* is very small, there are fewer individuals in the center of the release and more individuals near the outer side of the circle. This allows the drive to expand against the surrounding wild-type individuals more easily. Even if the drive is below its threshold in the center of the release area, the larger surrounding region with many drive individuals will have an advantage in expanding toward the very middle of the arena, making it less important for the drive to be initially released in the middle.

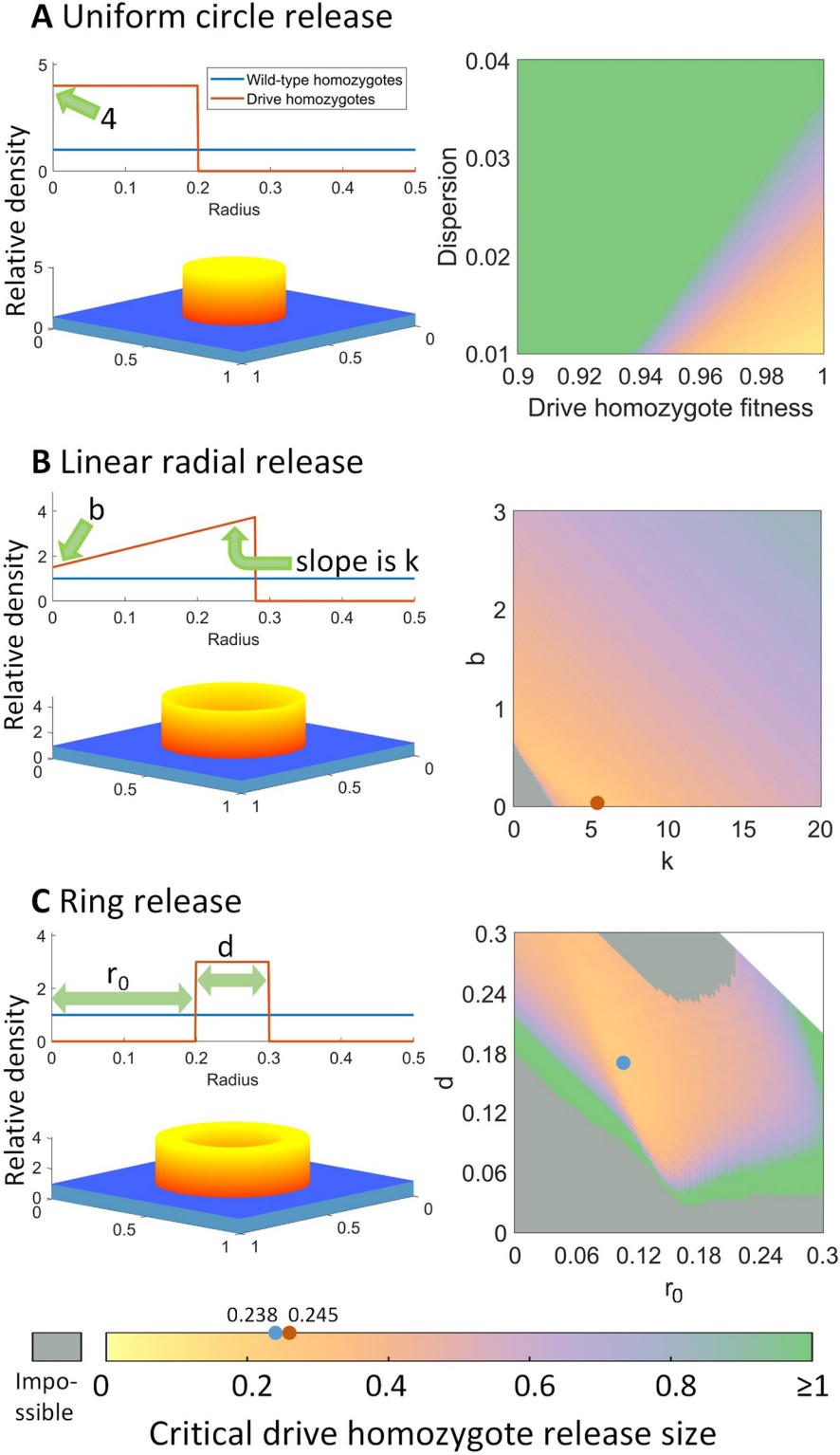

**Fig 5. Spatial circular release scenarios for the CifAB drive.** The heatmaps show the critical release size, which is the number of individuals required (expressed as a multiple of the carrying capacity of one unit area) for the drive to successfully establish and spread. Dots mark minimum critical release sizes. **A**: A fixed release frequency of 80% in the release area. As the critical release size increases, the release radius increases to support the greater number of drive individuals. **B**: A radial release pattern with varying density, keeping fitness at 1 and dispersion at 0.03. The maximum

radius in which individuals are released expands to accommodate more drive individuals. **C**: A ring release pattern with variable inner radius and width, keeping fitness at 1 and dispersion at 0.03. The area of the release is fixed, so a higher critical release size represents a higher drive release density. In the heatmaps, gray indicates that the drive is unable to spread under our assumptions (the drive will be lost with any release size that fits in the arena), and white requires a larger arena to be possible.

Inspired by the optimal result when $b$ is small, we also modeled a ring release that would allow us to completely avoid releasing drive individuals near the center of the arena. (Fig 5C), We collected the critical drive introduction density $h$ for each set of $r_0$ and $d$, thus calculating $V_c$, the critical relative release size. The optimal $V_c$ for the ring release is 0.238, when $r_0 = 0.104$, $d = 0.172$, $h = 0.774$, which is quite close to the critical release size of 0.245 for the linear radial release. Both the inner radius $r_0$ and ring width $d$ are moderate to ensure that the release density is not too high, which could cause rapid drive mortality due to competition from greatly exceeding the local carrying density.

## Self-limiting variants

Self-limiting drives can spread in a population for only a limited amount of time and then be eliminated due to intrinsic characteristics. Based on our 1-deme panmictic mathematical model, we examine some self-limiting variants of the CifAB drive, where *cifA* and *cifB* are in different genetic constructs. In these scenarios, because *cifA* may increase in frequency, we assume that this is where the drive's "cargo" will be and therefore its fitness cost. In our *Drosophila* model, no offspring are created if the male has both *cifA* and *cifB*, and the female has no *cifA* allele [58–60]. In the *Anopheles gambiae* model, only *cifB* is required in males to impose the toxin effect, so that when a male with only *cifB* mates with a female without *cifA*, they will also have no offspring [61].

The self-limiting properties of this drive are similar to killer-rescue systems. In these self-limiting scenarios, *cifB* can be viewed as the killer gene, and *cifA* is the rescue (Fig 6A). Males with *cifB* have no advantage over males without *cifB*, regardless of the population genetic structure or whether both *cifA* and *cifB* are required to make the toxin. In fact, they have a variable disadvantage by more often failing to reproduce due to toxin effects. Therefore, *cifB* allele will eventually be lost unless *cifA* is fixed. As long as the frequency of *cifB* is high enough, females with *cifA* are at an advantage since they can have offspring with any male. However, females without *cifA* cannot have offspring with males with the toxin. After *cifB* is lost (or reduced to a sufficiently low frequency), females with *cifA* no longer have an advantage. If *cifA* has a fitness cost, it is then bound to eventually be lost in the population. Here, we show trajectories of *cifA* and *cifB* drive allele and carrier frequencies as an example of the spread of self-limiting drives (Fig 6B).

In each variant, we are mainly interested in the protection time, which is the total number of generations *cifA* carrier frequency $\geq$ 80%. We also collect the average allele and carrier frequencies, maximum carrier frequencies for *cifA*, and the generation in which it reaches its maximum (S6–S9 Figs).

**1-locus 2-allele drives.** In this scenario, *cifA*, *cifB*, and wild-type are three different alleles that share the same locus. Denoting the alleles as $w$, $a$, $b$, we can get six combinations of genotypes: *ww*, *wa*, *wb*, *aa*, *bb*, *ab*. We introduce *cifA/cifB* heterozygotes (*ab*) at the introduction frequency $I$ and adjust *cifA* fitness along with the introduction frequency. We find that drive performance is limited when releasing heterozygotes when both *cifA* and *cifB* are required for the toxin effect (Fig 6C), and it requires a high introduction frequency and a very low fitness cost for the drive to persist in the population. In cases where only *cifB* is sufficient to cause the toxin effect, drive performance is increased but still rather limited (S5(B) Fig).

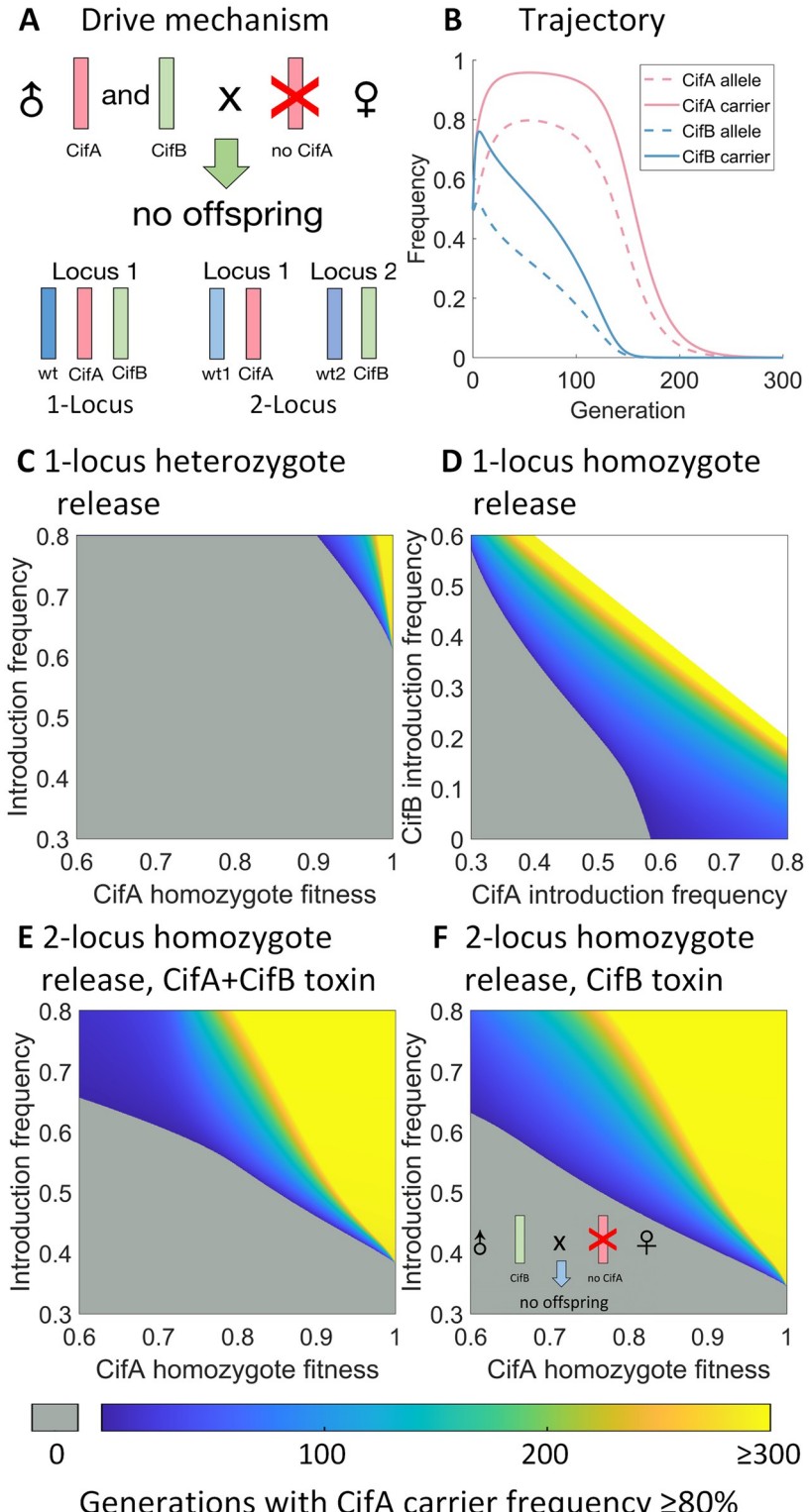

**Fig 6. Characteristics of self-limiting variants.** All homozygous fitness costs are on *cifA* allele and are 0.95 unless otherwise specified. **A**: When male *cifA* and *cifB* carriers mate with females lacking *cifA*, no viable offspring is produced (B-E). We model scenarios where *cifA* and *cifB* alleles must share the same locus (C-D) or are on separate genetically unlinked loci (B,E-F). **B**: Trajectories of allele and carrier frequencies. We use a homozygote introduction frequency of 0.5. **C**: A 1-locus *cifA*/*cifB* heterozygote release. **D**: A 1-locus homozygote release. **E**: A 2-locus

homozygote release. **F**: A 2-locus homozygote release, and only *cifB* is needed for the toxin effect. Gray indicates that the drive carrier frequency is unable to reach 80%. White indicates regions of parameter space that are impossible (the total fraction of starting drive individuals cannot exceed 1).

We also model scenarios where instead of heterozygotes, homozygotes of genotype *cifA/cifA* or *cifB/cifB* are introduced. When *cifA* homozygotes and *cifB* homozygotes are introduced at the same frequency, the performance of our gene drive is very similar to heterozygotes release scenarios and the drive is extremely confined (S5(A) Fig). To ascertain whether the ratio of *cifA* and *cifB* that is introduced into the population has an effect on drive performance, we introduced *cifA* homozygotes at introduction frequency $I_1$ and *cifB* homozygotes at $I_2$ (Fig 6D). We vary *cifA* introduction frequency from 0.3 to 0.8 and *cifB* introduction frequency from 0 to 0.6. We can conclude that the ratio of *cifA* and *cifB* only has a small effect on drive performance, which is mainly affected by the total introduction frequency. This is because higher *cifB* frequencies can result in a larger increase of *cifA* frequency over the first several generations before *cifB* declines to a low level.

**2-locus drives.** Often, underdominance gene drive systems are more robust when the different alleles are on different loci [19, 20, 33]. Thus, we propose introducing *cifA* and *cifB* onto different, genetically unlinked loci. The alleles on locus 1 are $w_1$ and $a$, which stand for wild-type and *cifA*. Similarly, the alleles on locus 2 are $w_2$ and $b$. We introduce double homozygotes with genotype *aa/bb* into a wild-type population. We apply this model to both possible cytoplasmic incompatibility rules, in which both *cifA* and *cifB* are required for the toxin effect as in Fig 6B (Fig 6E) and where *cifB* is sufficient to make the toxin (Fig 6F). We find that both models give broadly similar results. When the *cifA* homozygote fitness $\geq 0.9$, it is possible that the drive carrier frequency persists above 80% for a long period of time (over 300 generations) with a sufficient release size. However, when *cifA* fitness is relatively low $0.6 \leq F \leq 0.7$, the drive persists for longer at high frequency when only *cifB* is needed for the toxin. This is likely because when only *cifB* is needed to make the toxin, it's easier for the toxin effect to be activated, and in these additional cases of toxin activation, *cifA* alleles from males are not removed.

## Stochasticity of critical parameters

Deterministic, mathematical models are essentially estimates of results when population size $N \to +\infty$. However, in real-world scenarios, finite population sizes mean that results would be prone to the effect of stochasticity, particularly at lower population sizes. In some cases, this does not have a large effect on population dynamics, but for certain critical parameters, it may be important to understand the difference in possibilities inherent in stochastic systems.

Using our individual-based model in the panmictic population scenario for the CifAB drive without fitness costs, the actual drive establishment rate for different introduction frequencies varies with population size except when the release size is exactly equal to the introduction frequency, in which case it is always 50% (Fig 7A). Larger population sizes have steeper curves because stochastic effects (which could propel a drive above or below the threshold despite its starting frequency) play a smaller role. Therefore, the threshold is not so distinct when the population size is small. In such cases, the drive allele may still be lost at a considerable probability even if the introduction frequency exceeds the threshold, implicating that it is necessary to ensure that the introduction frequency is well above the threshold in these cases for a release to have guaranteed success. Similarly, if drive confinement is desired, it is essential for the drive to remain well below the threshold.

We compare our stochastic results with the results from our math models. The continuous-generation mathematical model shows that the introduction frequency threshold is 36.051%,

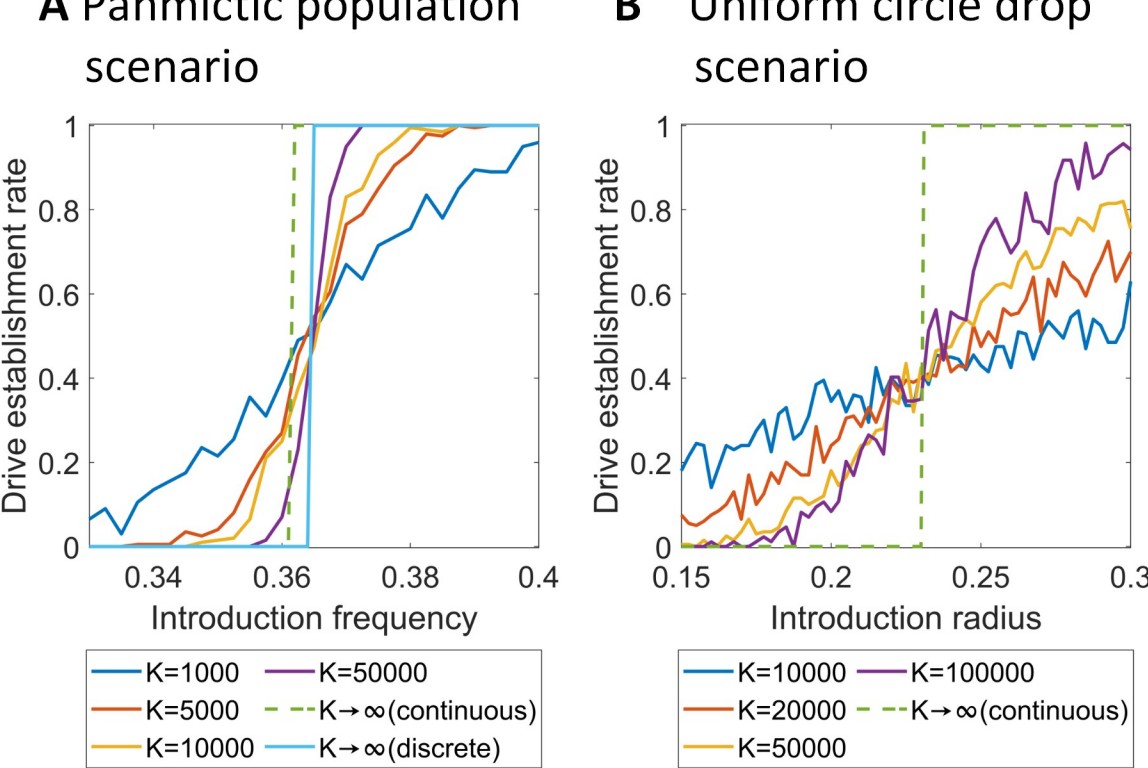

**Fig 7. Effect of population size on critical drive thresholds.** Results from simulations are shown, as well as results from continuous or discrete-generation math models. Drive fitness and efficiency are 1. **A**: Probability drive allele will be successfully established in a 1-deme panmictic model. **B**: Probability drive allele will successfully establish in a uniform circular 80% release in continuous space with a dispersion of 0.03. Simulations are repeated 200 times for each set of parameters.

which is lower than our panmictic model threshold. We also calculate the results using a discrete-generation math model and find that the threshold is 36.455%, exactly matching our discrete-generation simulation model. Through the comparison, we can infer that when $K \rightarrow +\infty$, the differences in continuous and discrete mathematical models are caused by initial effects. In fact, in the first few generations, drive allele frequency will slightly decrease and drive carrier frequency will increase in both discrete and continuous models (S4(B) Fig). The initial dip is milder in continuous models, but in discrete models, it might decrease too much in the first few generations to retain the capacity to later increase in frequency.

Stochastic effects play an important role in spatial scenarios as well. In these cases, stochastic effects can be even larger on a local scale where the number of interacting individuals is much smaller, even if the total population size is high. We considered a uniform circle release scenario and measured the rate at which the drive would successfully be established (Fig 7B). To observe stochastic effects in spatial scenarios, we adjust the carrying density of the population, resulting in different equilibrium population sizes. With the individual dispersion factor $v$ kept at 0.03 and the drive having no fitness cost, drive establishment in the spatial scenario is based on the introduction radius with a fixed introduction frequency of 0.8. As the total population size increases from 10000 to 100000, the slope of the success rate gets steeper, although local spatial effects still tend to keep this system more prone to stochastic effects than the panmictic scenario. Note that $K \rightarrow \infty$ for the spatial discrete model with non-overlapping generations is not available because the reaction-diffusion model used continuous time. However, we

can compare simulation data with data from our spatial continuous model. In our continuous-generation spatial mathematical model, the critical radius is 0.2305, which matches the intersection point of the curves when the population size is varied (unlike in the panmictic introduction threshold). However, this may be coincidental because the curves converge where the success rate is 40% rather than 50% as expected. The 40% convergence is likely because of our asymmetric competition function, which may interact with stochastic effects to produce different critical radii (with 50% success by definition) at different population densities.

## Discussion

Gene drives have been demonstrated in cage tests with *Anopheles* mosquitoes and flies, and they have also been modeled mathematically and computationally in different scenarios. However, they have not been deployed thus far due to a combination of technical challenges and various safety and sociopolitical concerns. A featured characteristic of our proposed *cifA*/*cifB* drives is that they are highly confined, at least in our mathematical models. We also modeled self-limiting variants, showing that it is possible to use the *cifA* and *cifB* alleles to make drives that are confined in time. Therefore, they could potentially be more safely tested and applied than drives with a zero introduction threshold such as homing drives.

Our study suggests that the CifAB drive can be considered strongly confined because in scenarios where *cifA* and *cifB* are in the same genetic construct and efficiency is fixed to 1, its introduction threshold ranges from 36% to 60%, according to the drive's fitness cost. These high thresholds could still allow the drive to spread within a continuous population (when fitness costs are sufficiently low) with a modest initial effort, but will greatly hamper the drive's spread to all but the most highly linked populations. Other factors such as dominance coefficient, drive element efficiency, and migration may have a substantial impact on drive performance and should be carefully considered because they will affect both confinement and the necessary effort for a successful release.

When we extend our model to 2D space and evaluate the drive wave of advance in space, we find that the minimum fitness needed for the drive wave to form ranges from 0.9 to 0.93. If a drive wave fails to form, then usually the drive will eventually be eliminated unless it is already present everywhere in a connected population. This fitness range is a lot higher than the range that allows a panmictic introduction threshold, showing unique properties of the drive in space and demonstrating stringent confinement. It is likely that any substantial obstacle, such as a migration corridor, would prevent the drive from spreading, even in directly connected populations [33]. Moreover, in spatial release scenarios, we found that the shape of drive release greatly impacts the effort needed for drive establishment. Though we did not likely find the optimal shape in this limited study, we saw that a ring-like drop with no individuals in the center is optimal for the shapes we considered. This result is likely broadly applicable to the release of confined gene drives and of *Wolbachia* and could thus potentially inform release programs.

Our proposed CifAB drive is closely related to *Wolbachia* for population modification strategies. They both perform reproduction manipulation through cytoplasmic incompatibility to spread through populations, and both are confined systems. However, the CifAB drive also possesses some unique characteristics. First, the mechanism of inheritance of the CifAB drive differs from that of *Wolbachia*. While *Wolbachia* are maternally inherited, the inheritance of CifAB drive follows Mendelian laws of segregation and independent assortment. This means that CifAB drive will have a moderate introduction threshold in idealized form due to loss of drive alleles in drive males that mate with wild-type females. In contrast, no *Wolbachia* are lost in the next generation due to failed mating between *Wolbachia* males and uninfected females. This means that *Wolbachia* only gains an introduction threshold if it is not an ideal drive in

terms of maternal transmission efficiency and fitness costs. What's more, the fitness costs of the CifAB drive may depend heavily on the cargo gene, which we introduce together with the drive allele. We can link any cargo to a CifAB drive, even a CRISPR cargo that targets and disrupts other genes. This makes the CifAB drive more flexible than *Wolbachia*, which currently cannot be genetically modified. *Wolbachia* has the advantage of itself being able to block transmission of pathogens [52, 53, 74]. It has a lower threshold than the CifAB drive, but in practice, it will usually carry substantial fitness costs, which reduces the magnitude of the difference between introduction thresholds. Depending on the species, it may be easier to modify its genome than to infect it with *Wolbachia*, or at least *Wolbachia* that are incompatible with those that can already be found in a species. CifAB could also be more easily used to modify a population that is infected with a different (and perhaps unwanted) *Wolbachia* strain because a new *Wolbachia* strain in this instance would have a 50% introduction threshold if equal to the existing strain.

CifAB drive also has similarities to previously proposed gene drives, in particular toxin-antidote underdominance drives. Though not an underdominance drive itself (drive heterozygotes and homozygotes have similar reproductive success), both CifAB drive and underdominance drives have introduction thresholds even without any fitness costs (other than those directly related to the drive mechanism). CifAB has an introduction frequency threshold near 36%, while other underdominance drives have a variable threshold between 18% and 67% (or even higher for certain suppression drives) [19, 20, 23, 33]. Other non-underdominance toxin-antidote drives have a threshold of zero, but one will appear with any fitness cost [26]. Therefore, the CifAB drive is a medium-threshold drive compared to the others, being strongly confined in space but still being able to form a wave of advance when fitness costs are low. Several strategies have been proposed for making confined gene drives, such as RNAi [20, 23, 33], incompatibilities [28], and CRISPR [19, 26], so *Wolbachia* phage genes provide an increased diversity of options. Each of these mechanisms has potential advantages or disadvantages depending on the situation. CifAB drives can potentially have an advantage because they don't rely on specific target sequences like CRISPR or RNAi that could contain resistant sequences. Alike with other confined drives, the CifAB drive needs a suitable promoter to ensure high drive performance [59, 60]. The promoter could potentially even change the drive's dynamics, such as a *cifA* promoter that could be zygotically expressed and still serve as a rescue to the toxin effect. This could substantially reduce the drive's threshold, though it is unclear if this would be possible with a CifAB drive. Another advantage of the CifAB drive is that different forms of *cifA* and *cifB* can potentially be used to easily replace one drive with another, much like *Wolbachia* with different cytoplasmic incompatibility [63, 75]. This is possible with other drives too, but requires more complicated engineering.

When *cifA* and *cifB* are in different genetic constructs, the drive is self-limiting in time and will be eliminated from the population eventually as long as the *cifA* allele bears a fitness cost. This makes the variant drive similar to killer-rescue drives [35, 37], split drives [38, 40, 76, 77], and daisy-chain gene drives [36, 78], which are also self-limiting. The *cifA*/*cifB* drive could potentially be easier to engineer than these other systems, which tend to involve complex mechanisms or many different alleles, though more experience with self-limiting drives in non-model organisms is needed before this can be confidently assessed. Nevertheless, *cifA*/*cifB* drive would certainly be easier to control than split homing drives or daisy-chain drives, which are highly sensitive to release size [36, 79]. They may also offer potentially greater persistence than killer-rescue drive [35, 41]. Indeed, under a range of parameters, the self-limiting 2-locus *cifA*/*cifB* drive lasted for several hundred generations, which is likely longer than a cargo gene could last if the cargo is prone to inactivating mutation that eliminates its fitness cost. This could make self-limiting *cifA*/*cifB* drives a particularly desirable tool for long-term modification strategies that still eventually disappear from wild populations.

Our models were necessarily simplified to present an initial introduction to the basic properties of these new drives, and indeed, further work could provide a more fundamental understanding if analytical solutions could be obtained. In our mathematical model, generations were continuous, but one limitation is that the result was deterministic. All the calculations were done according to the fraction of individuals of each genotype relative to carrying capacity under the assumption that the size of the population $N \rightarrow +\infty$. In real populations with limited size, the outcome may be affected by stochastic fluctuations. In contrast, the simulation model is somewhat complementary. It is individual-based, and could show the effects of stochasticity at small population sizes, while generally recapitulating the results of the mathematical model when the population size is very large. However, generations are discrete, and all individuals mate at the same time in each generation, which differs from real-life scenarios where time is continuous and the life cycles of different individuals might be staggered.

Both models omitted the influence of several factors that may have an effect on the outcome of the drive. For example, we ignored the impact of interspecies competition and assumed that the carrying capacity *K* is constant over time when many real-world environments change with seasons. Our models are simple representations of general populations, lacking species-specific life cycle details. We make the assumption that males and females have equal survival, but in some species such as mosquitoes, females may survive for longer. To model a target species more realistically, it is also necessary to consider seasonal population fluctuations, competitors, predators, prey, survival characteristics, age structure, and the effect of mutations. In our spatial models, we assumed that the terrain was flat and homogeneous, and that the dispersion of individuals across the terrain was random in all directions, which are both simplifications. These factors could be investigated in future species-specific studies.

Because *Wolbachia* can be found broadly in many different species of insects, drives based on *cifA*/*cifB* and similar genes could potentially function in a wide variety of insect species, including many mosquito species that are important disease vectors. However, the construction of such drives still requires consideration of many practical issues. First, it's imperative that we have good control over the expression of *cifA* and *cifB*, whether they're in the same genetic construct or not [59, 61]. High expression of *cifA* and *cifB* (or possibly just *cifB* in some cases) is needed in males so that the toxin effect can manifest in most offspring. Therefore, to ensure that the toxin effect is experienced by offspring, a strong male germline promoter is required. Even more important is ensuring sufficient *cifA* expression in females, so that the antidote effect could reliably counter the toxin and make offspring viable. Lastly, to minimize the fitness cost and lower the introduction frequency threshold, it may be necessary to choose promoters that minimize overexpression of drive alleles and undesired expression in somatic cells. Thus far, constructs with these genes have been successfully constructed in *Drosophila melanogaster* [59] and *Anopheles gambiae* [61], though gene drive scenarios and associated fitness effects have not been evaluated in these or in other species.

Overall, we have found that gene drives based on *Wolbachia* phage genes involved in cytoplasmic incompatibility have promising properties in simulation and mathematical models. Constructs with *cifA* and *cifB* could be made into new drives that are confined and effective with potential use in many insect species. Further studies are needed to investigate the potential of these drives in various species and in more realistic modeling scenarios.

## Supporting information

**S1 Fig. Basic characterization of the CifAB drive. A**: The introduction threshold as a function of drive homozygote fitness. The drive allele is lost for any introduction frequency below

1 when fitness is less than 0.72. **B,C**: Average drive allele and carrier frequencies in the first 100 generations after releasing drives for each set of parameters.
(TIF)

**S2 Fig. Effect of drive efficiency and dominance coefficient. A**: The introduction threshold as a function of drive efficiency, which is the value of both toxin and antidote efficiency. The CifAB allele is lost for any introduction frequency below 1. **B,C**: Average drive allele and carrier frequencies in the first 100 generations after releasing drives for each set of parameters.
(TIF)

**S3 Fig. Generation drive carrier frequency reaches 70% in a 2-deme model.** In 2-deme scenarios, the first generation when drive carrier frequency reaches 70% is collected in **A**: introduction deme and **B**: linked deme. Gray indicates that drive is lost or carrier frequency is unable to reach 70% in the first 50 generations after drive release.
(TIF)

**S4 Fig. Comparison between mathematical and simulation models. A**: Wave speed measurements in spatial math and simulation models essentially match when dispersion factor $v$ is below 0.07. The wave speed in the models has a slight difference when dispersion is increased due to stochastic effects (a larger region of low drive frequency at higher dispersion increases this effect). **B**: Frequency trajectories of the continuous and discrete-generation mathematical models with the same release conditions, showing an example where results substantially differ between the models near critical release values. This is because in the first few generations, allele frequency may decrease more in the discrete-generation model, making the drive allele more likely to get lost in the population.
(TIF)

**S5 Fig. Additional self-limiting scenarios.** We model scenarios where *cifA* and *cifB* alleles must share the same locus. **A**: A 1-locus homozygote release with both *cifA* and *cifB* is required for the toxin. Both *cifA*/*cifA* homozygotes and *cifB*/*cifB* homozygotes are introduced at the specified introduction frequency. **B**: A 1-locus *cifA*/*cifB* heterozygote release, and only *cifB* is needed for the toxin effect. Gray indicates that the drive carrier frequency is unable to reach 80%.
(TIF)

**S6 Fig. Average *cifA* allele frequency in self-limiting drive scenarios.** All homozygous fitness costs are on the *cifA* allele and are 0.95 unless otherwise specified. **A**: A 1-locus *cifA*/*cifB* heterozygote release. **B**: A 1-locus *cifA*/*cifB* heterozygote release, and only *cifB* is needed for the toxin effect. **C**: A 1-locus homozygote release. Both *cifA*/*cifA* homozygotes and *cifB*/*cifB* homozygotes are introduced at introduction frequency. **D**: A 1-locus homozygote release. **E**: A 2-locus homozygote release. **F**: A 2-locus homozygote release, and only *cifB* is needed for the toxin effect. White indicates regions of parameter space that are impossible (the total fraction of starting drive individuals cannot exceed 1).
(TIF)

**S7 Fig. Average *cifA* carrier frequency in self-limiting drive scenarios.** All homozygous fitness costs are on the *cifA* allele and are 0.95 unless otherwise specified. **A**: A 1-locus *cifA*/*cifB* heterozygote release. **B**: A 1-locus *cifA*/*cifB* heterozygote release, and only *cifB* is needed for the toxin effect. **C**: A 1-locus homozygote release. Both *cifA*/*cifA* homozygotes and *cifB*/*cifB* homozygotes are introduced at introduction frequency. **D**: A 1-locus homozygote release. **E**: A 2-locus homozygote release. **F**: A 2-locus homozygote release, and only *cifB* is needed for the toxin effect. White indicates regions of parameter space that are impossible (the total fraction

of starting drive individuals cannot exceed 1).
(TIF)

**S8 Fig. Maximum *cifA* carrier frequency in self-limiting drive scenarios.** All homozygous fitness costs are on the *cifA* allele and are 0.95 unless otherwise specified. **A**: A 1-locus *cifA*/*cifB* heterozygote release. **B**: A 1-locus *cifA*/*cifB* heterozygote release, and only *cifB* is needed for the toxin effect. **C**: A 1-locus homozygote release. Both *cifA*/*cifA* homozygotes and *cifB*/*cifB* homozygotes are introduced at introduction frequency. **D**: A 1-locus homozygote release. **E**: A 2-locus homozygote release. **F**: A 2-locus homozygote release, and only *cifB* is needed for the toxin effect. White indicates regions of parameter space that are impossible (the total fraction of starting drive individuals cannot exceed 1).
(TIF)

**S9 Fig. Generation when *cifA* carrier frequency reaches its maximum in self-limiting drive scenarios.** All homozygous fitness costs are on the *cifA* allele and are 0.95 unless otherwise specified. **A**: A 1-locus *cifA*/*cifB* heterozygote release. **B**: A 1-locus *cifA*/*cifB* heterozygote release, and only *cifB* is needed for the toxin effect. **C**: A 1-locus homozygote release. Both *cifA*/*cifA* homozygotes and *cifB*/*cifB* homozygotes are introduced at introduction frequency. **D**: A 1-locus homozygote release. **E**: A 2-locus homozygote release. **F**: A 2-locus homozygote release, and only *cifB* is needed for the toxin effect. White indicates regions of parameter space that are impossible (the total fraction of starting drive individuals cannot exceed 1).
(TIF)

## Author Contributions

**Conceptualization:** Jiahe Li, Jackson Champer.

**Data curation:** Jiahe Li, Jackson Champer.

**Formal analysis:** Jiahe Li.

**Funding acquisition:** Jackson Champer.

**Investigation:** Jiahe Li.

**Methodology:** Jiahe Li.

**Project administration:** Jackson Champer.

**Resources:** Jiahe Li.

**Software:** Jiahe Li.

**Supervision:** Jackson Champer.

**Validation:** Jiahe Li.

**Visualization:** Jiahe Li.

**Writing – original draft:** Jiahe Li, Jackson Champer.

**Writing – review & editing:** Jiahe Li, Jackson Champer.

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
