## [Decision Letter · Decision Letter 0]

22 Nov 2022

Dear Dr Champer,

Thank you very much for submitting your Research Article entitled 'Harnessing Wolbachia cytoplasmic incompatibility alleles for confined gene drive: a modeling study' to PLOS Genetics.

The manuscript was fully evaluated at the editorial level and by independent peer reviewers. The reviewers appreciated the attention to an important topic but identified some concerns that we ask you address in a revised manuscript. We therefore ask you to modify the manuscript according to the review recommendations. Your revisions should address the specific points made by each reviewer.

Yours sincerely,

Gregory P. Copenhaver

Editor-in-Chief

PLOS Genetics

Reviewer's Responses to Questions

**Comments to the Authors:**

Reviewer #1: “Harnessing Wolbachia cytoplasmic incompatibility alleles for confined gene drive: a modeling study” builds off three important advances from recent years: the expansive literature on synthetic drive systems, Wolbachia’s deployment as a tool in vector control worldwide (reliant on CI and pathogen interference traits), and the discovery of CI’s genetic basis. CI (or cytoplasmic incompatibility) is a conditional sterility where embryos die when infected males fertilize uninfected eggs but not infected eggs.

This study mathematically assess how Wolbachia’s CI genes (cifA and cifB) could be used as a gene drive system encoded in the host genome. In this scenario, the cifAB drive would be paired with a payload that confers resistance to a disease of interest and then spreads that resistance into a vector population. I don’t have a complete grasp of the mathematics underlying their analyses and cannot effectively comment on their accuracy. However, I am confident that the broader principles of their analyses are well reasoned, and their analyses are thorough, with some gaps. Below, I highlight three larger issues and provide some specific line-by-line feedback.

1) Literature review – The literature review for this paper could be vastly improved. This is particularly relevant to the Wolbachia literature from which the foundations of this paper are manifest. In the line-by-line feedback I provide some specific recommendations for papers to cite. However, I would encourage the authors to broadly review the literature on CI as a mechanism for spread, Wolbachia as a tool for vector control, and CI’s mechanistic underpinnings. For instance, surprisingly missing, but highly relevant, is a 1999 study from Turelli and Hoffmann in Insect Molecular Biology. They also model how CI transgenes in the host nuclear genome could be used for drive but of course do this without knowing the specific details of the CI genes.

2) Comparisons to cytoplasmic drive – This manuscript provides a rather thorough investigation of the parameter space for a transgenic cifAB gene drive. However, no effort is made to formally compare this drive mechanism to the natural cytoplasmic drive. Given that the cytoplasmic drive is currently being deployed and protects millions from disease, it makes sense that cifAB gene drive should not only be compared to other gene drive systems but also to the “wild-type” system it would aim to replace or complement. This could be addressed by adding a description of the ways that the models for spread differ when the cif genes are encoded in the host vs the Wolbachia genome and adding analyses where the cifs are Wolbachia encoded.

3) Fitness advantage – Numerous analyses consider the impacts of the relative fitness of the drive strain relative to the wild type. However, these studies assume that the drive strain has an equal or lower fitness than the wild type. While this assumption is reasonable in many scenarios, it remains plausible that the drive strain could have a fitness advantage that would considerably impact the spread dynamics and have implications for its role as a confined gene drive. There are cases in the text where this is described verbally, but the analyses could be improved by displaying a fitness advantage scenario.

Line-by-line feedback:

Abstract sentence 1 – ‘Wolbachia’ is plural.

Abstract sentence 3 – ‘alternative mechanism’ would be more accurate as ‘in other systems’. It’s still debated whether it’s truly an alternative mechanism.

Line 120 – I recommend removing reference 42 and adding Hoffmann 1990, Kriesner 2013, and Kriesner 2016 that describe the parameters that govern Wolbachia’s spread.

Line 141 – The work from Verily should also be cited here. For example, Crawford et al 2020 and Beebe et al 2021. Both focus on IIT in Aedes aegypti.

Line 162 – The first letter of the Wolbachia strain name should be italics. For instance, in wMel, the w is italics.

Line 163 – Utarini et al. 2021 should be cited. They show that Wolbachia releases in Yogyakarta inhibit local dengue spread by over 85%.

Line 165 – It’s more accurate to say “a pair of genes commonly associated with phage WO.” There are rare exceptions where they are not in the phage.

Line 168 – Specifically, Drosophila melanogaster. Also, the way this is written needs to be revised slightly since it implies that whether one or both genes are required is host dependent, but it’s more complex than that. For instance, Sun et al. 2022 showed CifB from wNo Wolbachia is sufficient to cause transgenic CI in D. melanogaster. In reality it seems that CifA’s requirement in CI might be dependent on both Wolbachia and host genotypes. For simplicity it’s possible to say something like “transgenic expression of cifB is sometimes sufficient to cause CI, though co-expression of cifA and cifB is required in other systems”.

Line 172 – Reference 53 isn’t particularly relevant to this statement.

Line 176 – Needs a few sentences or paragraph describing the rescue side of things. Later it is mentioned that CifA rescues CI, but background isn’t provided. Shropshire et al. 2018 and 2019 show CifA rescues CI and Xiao et al. 2021 and Wang et al. 2022 suggest that CifA rescues CI by binding CifB.

Line 191 – A figure, or part of a figure, illustrating the different strategies could be helpful.

Line 193 – When referencing a gene, the first letter of the symbol should be lower case and the symbol italicized (e.g., cifA and cifB, in italics). As written (CifA and CifB) it implies a protein is being discussed. This is an error throughout the paper.

Line 209 – There is a robust literature from Turelli, Barton, and others on the parameters that govern Wolbachia’s spread. This paper seems an excellent opportunity to lay out the different assumptions underlying Wolbachia’s spread via CI and CifAB gene drive.

Line 289 – Cite literature on efficiency variation. Shropshire et al. 2020 eLife reviews the literature on this topic. Also, Ritchie et al 2022 looked at the impact of temperature and male age on transgenic CI.

Fig 1A – The drive mechanism also requires that females with the drive allele can rescue, correct? This should be illustrated here for completion.

Fig 1B – While the legend says that gray means the drive allele is eliminated, the key indicates that it will spread given > 20 generations. Is the legend for Fig 1B accurate given that it tops at 20, but Fig 1D indicates that some introduction frequencies/fitness consequences take far longer than that?

Fig 1C,D – Include the fitness value in the legend or figure. In fact, when it’s relevant, it would be helpful to describe the important fixed parameters in the figure or legend throughout the manuscript.

Line 739 – How many generations will it take for 36.051% to reach fixation? The figure shows it very slightly increasing by generation 40.

Fig 2 legend – “dive” should be “drive”

Fig 3 – For consistency with other figures, and to make it easier to read, can gray again be used to indicate parameter space where the drive allele fails to spread? The difficult part with using blue here is the statement in the legend that says “Blue colors tend to represent”. It’d be helpful to know where the line is between tend to not, and do.

Line 851 – It would be valuable to consider a scenario where instead of varying introduction frequency, equilibrium carrier/allele frequency is varied relative to migration. In this scenario I’m curious what would happen if there was a linked deme with a late-stage introduction deme with a stable drive frequency > 90%. What level of migration would be required from this stable population into the linked population to make the drive spread?

Line 917 – It would help to have context for what a wave speed of these magnitudes would mean in a natural context. It’s hard to assess this without a deeper comparison of the simulated arenas relative to a wild population.

Line 1048 – It’s relevant to cite literature from Turrelli (Turelli 1994, for example).

Line 1229 – While the math supports them being highly confined, empirical data are needed to support the math. As such, a bit more caution is warranted here and throughout. This is especially true given that a positive fitness effect is likely to prevent this from being self-limiting.

Lline 1270 – You first say “did not likely find the optimal shape” but then say “ring-like drop with no individuals in the center is optimal.”

Line 1280 – Another key difference is the mechanism of inheritance.

Line 1331 – Relevant to cite the transgenic CI literature regarding promoters successfully used to induce transgenic CI. For example, Shropshire et al. 2018 and 2019 show how promoter variation impacts rescue and CI respectively.

Line 1339 – Need to cite the bidirectional CI literature. For instance, Wang et al. 2022 show that binding site variation between cifA and cifB contributes to bidirectional CI. Shropshire et al. 2021 also shows how cifs from different strains interact transgenically.

Line 1440 – Notably, the systems used for transgenic CI have largely used germline promoters and thus should have minimal expression in somatic tissues.

Reviewer #2: Summary:

The authors propose new types of toxin-antidote non-suppression gene drive systems using Wolbachia CifA and/or CifB genes independent of the Wolbachia bacteria and without using CRISPR homing. They did a good job of describing its properties in detail, i.e. using deterministic/mathematical models with continuous time, stochastic simulations with discrete time, and spatially explicit deterministic and stochastic models.They showed that when introduced together, these two genes result in a confined or localised gene drive; while introduced separately the drive system becomes self-limiting or transient or temporary. They also showed that the manner of release significantly affects drive persistence in spatial context. This new drive system seems to be more feasible and potentially safer to test than homing CRISPR-based gene drives.

Major Comments:

I have no major comments doubting the mathematical models and stochastic modelling employed. As far as I can tell, the models and simulation methodologies are sound.

Minor Comments:

- In the abstract, it’s potentially helpful to explicitly specify that Wolbachia bacteria as whole organisms are not involved in the gene drive system - just their cytoplasmic incompatibility genes.

- Lines 212-217: I understand on a high-level that defining a generation in this way makes sense, but is it actually equivalent or similar to that of discrete non-overlapping generations? This potentially warps generation interval right after drive introduction as population size decreases as the fitnesses of wild-type females drop.

- Table 1: population size, N is not the “number” of individuals, rather the relative population size, i.e. number of individuals relative to the number of individuals at carrying capacity.

- Lines 587-595: What were the border conditions used: toroidal or absorbing or reflecting boundaries, etc?

- I think I understand that you used overlapping generations in your math models because it was a straightforward extension into spatially explicit models; however, did you bother with using discrete non-overlapping deterministic models (which is a better comparison with your stochastic simulations)? Would you expect similar properties?

- Figure 1 C and D: It’s probably advantageous to add vertical lines corresponding to when Hardy-Weinberg equilibrium (HWE) was reached after homozygous drive introduction so that the unstable equilibria do not look funny, i.e. kinks around the equilibria where we expect introduction frequencies equal to the equilibria to not change since they’re the equilibria.

- Figure 1 C and D: Also, does it make more sense to use a unitless time axis instead of “generations” because of generation-time warping with continuous generations?

- Additionally, is the effect of the drive on population size insignificant?

- Figure 6 B, E and F: Similar comment to Figure 1 C and D - express the x-axis as time instead of generations.

- Figure 7 B: note that K-> infinity for the spatial discrete non-overlapping generations is not available due to the reaction-diffusion model used continuous time.

Reviewer #3: This paper introduces a potential new approach to creating a gene drives for the control of insect disease vectors. The idea is to utilise the genes that cause the intracellular bacteria Wolbachia to spread in many arthropod species, by a mechanism called cytoplasmic incomparability, without the necessity of introducing the entire bacterial genome. It is an appealing prospect and clearly worthy of investigation. The paper does this using a suite of mathematical models that differ in complexity from non-spatial to spatial and deterministic to a spatial stochastic simulation. This is an excellent approach to exploring the roles of different complexities and I found the model progression logical and well explained. Overall, I enjoyed reading the manuscript; it is clearly written, instructive, and makes a good case for further research into the proposed technology. I only have a few minor suggestions for the authors to consider.

1. The figures were generally clear and visually appealing. In figure 1 panels B,E and F, I wondered if you could have changed the scaling to emphasise the variation – e.g. 1B is almost entirely yellow with a tiny bit of green – could you reassign the scaling so there is more green?

2. There is an unexplained white region in fig. 6D – I presume this is meant to be yellow? (or indefinite persistence – though this would also be yellow?).

3. You used numerical methods throughout to analyse the deterministic models (that you call “math models”), but I wondered if it might be possible to obtain analytical solutions. E.g. for the introduction threshold from the panmictic model – did you try? To be clear I think the analyses based on numerical results are insightful, yet analytical solutions might have given even a bit more insight into the interactions between model parameters.

4. A small typo on pg 14 line 994 – remove “is able”.

**Have all data underlying the figures and results presented in the manuscript been provided?**

Reviewer #1: Yes

Reviewer #2: Yes

Reviewer #3: Yes

PLOS authors have the option to publish the peer review history of their article (what does this mean?). If published, this will include your full peer review and any attached files.

Reviewer #1: No

Reviewer #2: **Yes: **Jefferson Paril

Reviewer #3: No

---

## [Decision Letter · Decision Letter 1]

21 Dec 2022

Dear Dr Champer,

We are pleased to inform you that your manuscript entitled "Harnessing Wolbachia cytoplasmic incompatibility alleles for confined gene drive: a modeling study" has been editorially accepted for publication in PLOS Genetics. Congratulations!

Yours sincerely,

Gregory P. Copenhaver

Editor-in-Chief

PLOS Genetics

Comments from the reviewers (if applicable):

Reviewer's Responses to Questions

**Comments to the Authors:**

Reviewer #1: I was reviewer 1 in the first round of revision. The authors have adequately addressed my concerns and provided robust rationale when they disagreed with my opinions. I have no additional comments. Congratulations to the authors for an excellent article.

Reviewer #2: The authors addressed all the issues I have listed in the review in a satisfactory manner. However, I would still suggest that they include a statement for Figure 1 indicating that equilibria were not reached for the seemingly flattening out trajectories between loss and fixation frequencies.

**Have all data underlying the figures and results presented in the manuscript been provided?**

Reviewer #1: Yes

Reviewer #2: Yes

PLOS authors have the option to publish the peer review history of their article (what does this mean?). If published, this will include your full peer review and any attached files.

Reviewer #1: No

Reviewer #2: No

**Data Deposition**

http://datadryad.org/submit?journalID=pgenetics&manu=PGENETICS-D-22-01086R1

**Press Queries**

---

## [Editor Report · Acceptance letter]

17 Jan 2023

PGENETICS-D-22-01086R1 

Harnessing Wolbachia cytoplasmic incompatibility alleles for confined gene drive: a modeling study 

Dear Dr Champer, 

We are pleased to inform you that your manuscript entitled "Harnessing Wolbachia cytoplasmic incompatibility alleles for confined gene drive: a modeling study" has been formally accepted for publication in PLOS Genetics! Your manuscript is now with our production department and you will be notified of the publication date in due course.

With kind regards,

Anita Estes

PLOS Genetics

On behalf of:
